# CAUTIOUS WEIGHT DECAY

**Lizhang Chen**[*,1,2]    **Jonathan Li**[*,1]    **Kaizhao Liang**[1]    **Baiyu Su**[1]    **Cong Xie**
**Nuo Wang Pierse**[2]    **Chen Liang**[2]    **Ni Lao**[1,2]    **Qiang Liu**[1]
[1]The University of Texas at Austin    [2]Google LLC, Mountain View, California, USA

## ABSTRACT

We introduce Cautious Weight Decay (CWD), a one-line, optimizer-agnostic modification that applies weight decay only to parameter coordinates whose signs align with the optimizer update. Unlike standard decoupled decay, which implicitly optimizes a regularized or constrained objective, CWD preserves the original loss and admits a bilevel interpretation: it induces sliding-mode behavior upon reaching the stationary manifold, allowing it to search for locally Pareto-optimal stationary points of the unmodified objective. In practice, CWD is a drop-in change for optimizers such as ADAMW, LION, and MUON, requiring no new hyperparameters or additional tuning. For language model pre-training and ImageNet classification, CWD consistently improves final loss and accuracy at million- to billion-parameter scales.

## 1  INTRODUCTION

---

**Algorithm 1**  Cautious Weight Decay (CWD)

---

**given** parameters $\mathbf{x}_t$, optimizer update $\mathbf{u}_t$, learning rates $\eta_t > 0$, weight decay coefficient $\lambda \geq 0$

$\mathbf{x}_{t+1} \leftarrow \mathbf{x}_t - \eta_t \left( \mathbf{u}_t + \lambda \mathbb{I}(\mathbf{u}_t \mathbf{x}_t \geq \mathbf{0}) \mathbf{x}_t \right)$           ▷ entrywise multiplication

---

**Optimization algorithms** lie at the core of modern deep learning, shaping not only convergence speed but also training stability and generalization ability across domains such as natural language processing and computer vision. As models and datasets scale, traditional methods such as stochastic gradient descent (SGD) and SGD with momentum (Sutskever et al., 2013) encounter limitations, including *slow convergence in non-convex landscapes, sensitivity to learning rate schedules, and poor robustness to sparse or noisy gradients* (Scaman & Malherbe, 2020; Zhao et al., 2025). In response, a wide range of alternatives have emerged, including adaptive gradient methods (Duchi et al., 2011; Kingma & Ba, 2015), approximate second-order approaches (Martens & Grosse, 2015; Gupta et al., 2018; Yao et al., 2021; Liu et al., 2024; Nguyen et al., 2024; Wen et al., 2025), and specialized algorithms for extreme training regimes (Liang et al., 2024b; Luo et al., 2024; Xie et al., 2024; Huang et al., 2025; Zhang et al., 2025).

Among these advances, **decoupled weight decay** (Loshchilov & Hutter, 2019) has proven especially influential. In its general form, decoupled weight decay augments any optimizer update $\mathbf{u}_t$ with a decay term applied directly to the parameters, i.e.

$$\mathbf{x}_{t+1} \leftarrow \mathbf{x}_t - \eta_t(\mathbf{u}_t + \lambda \mathbf{x}_t), \qquad \mathbf{u}_t = \text{OptimizerUpdate}(\mathbf{x}_t).$$

This technique improves training stability and generalization by preventing the adaptive learning rates from interfering with regularization, as exemplified by the success of ADAMW in large model training (Brown et al., 2020; Dosovitskiy et al., 2021; Touvron et al., 2023) and the subsequent development of state-of-the-art optimizers such as LION (Chen et al., 2023), LION-$\mathcal{K}$ (Chen et al., 2024), and MUON (Jordan et al., 2024; Liu et al., 2025).

However, decoupled weight decay *remains agnostic to the directional alignment between the optimizer update and the parameters*, which may hurt performance when they conflict. Intuitively, when

---

*Equal contribution by LC and JL. Correspondence: `lzchen, jli@cs.utexas.edu`

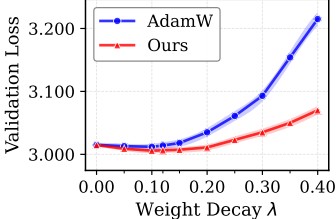 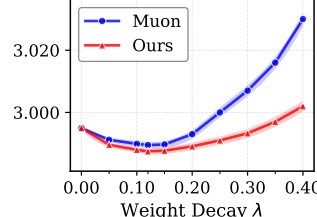 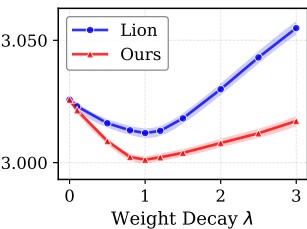

Figure 1: Final validation loss vs. weight decay coefficient $\lambda$ for 338M models trained on C4 under Chinchilla scaling. Our approach (red) achieves lower final loss than standard weight decay (blue) while preserving the optimizer-specific optimum in $\lambda$. For each optimizer (ADAMW, LION, MUON), both methods use the same hyperparameters.

the update $\mathbf{u}_t$ and parameters $\mathbf{x}_t$ point in the same direction for a given dimension, weight decay acts as a regularizer that improves stability; however, when their directions differ, applying decay actively resists beneficial movement toward the optimum. Furthermore, decoupled weight decay has been shown to implicitly impose regularization terms on the objective function (Chen et al., 2024; Xie & Li, 2024), which corresponds to parameter norm constraints for ADAMW, LION, and MUON.

In light of these limitations, we propose a simple refinement: cautious weight decay (CWD), in which decay is applied *only* in dimensions where the update and parameter signs align (Algorithm 1). Our main contributions are as follows.

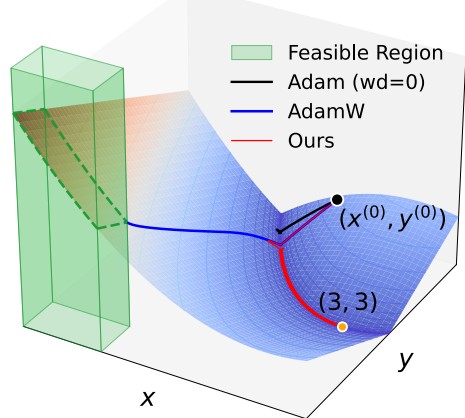

Figure 2: Trajectories of ADAM, ADAMW, and ADAM + CWD on a toy example. ADAM halts at a minimizer, while ADAMW minimizes the objective within a constrained region (green). In contrast, ADAM + CWD exhibits sliding mode dynamics within the minimizer manifold.

• We introduce cautious weight decay, a sign-selective extension of decoupled decay that applies weight decay only when the parameters and update align. Our technique can be implemented as a one-line modification without introducing additional hyperparameters compared to standard decoupled decay.

• We use Lyapunov analysis to show that standard optimizers (SGD(M), LION-$\mathcal{K}$, ADAM) with cautious weight decay are asymptotically stable and unbiased, in the sense that they optimize the original loss rather than a regularized surrogate. The regularization effect of cautious weight decay instead becomes a bilevel objective of finding locally Pareto-optimal points within the stationary manifold (Figure 2). Furthermore, we show a convergence rate for discrete-time ADAM with cautious weight decay in the smooth nonconvex setting under additional assumptions.

• In language modeling (OLMo et al., 2025; Kamath et al., 2025) and ImageNet classification (Deng et al., 2009), we observe that cautious weight decay generally accelerates convergence and lowers final validation loss for ADAMW, LION, and MUON (e.g., Figure 1). These improvements translate into higher zero-shot accuracy on standard benchmarks from 338M to 2B parameters and across architectures without retuning baseline settings ($\approx$20,000 NVIDIA H100 HBM3-80GB GPU hours for all experiments).

## 2 BACKGROUND AND MOTIVATION

### 2.1 DECOUPLED WEIGHT DECAY

Gradient-based optimizers with decoupled weight decay can be characterized by the update rule

$$\mathbf{x}_{t+1} = (1 - \eta_t\lambda)\mathbf{x}_t - \eta_t\mathbf{u}_t, \tag{1}$$

where $\mathbf{u}_t := \mathcal{U}(\mathbf{x}_t, \mathbf{g}_1, \ldots, \mathbf{g}_t, t)$ is an adaptive, often sign-normalized update vector constructed from first and second-moment estimates (e.g., momentum buffers, diagonal preconditioners), $\eta_t > 0$ is the learning rate, and $\lambda \geq 0$ is the decoupled weight decay coefficient. This framework encapsulates a wide range of standard optimizers for machine learning, including ADAMW and LION-$\mathcal{K}$.

**ADAMW.** The update vector is given by $\mathbf{u}_t = \mathbf{D}_t^{-1}\widehat{\mathbf{m}}_t$, where $\mathbf{D}_t$ is a diagonal preconditioner and $\widehat{\mathbf{m}}_t$ is bias-corrected first-moment estimate. Explicitly,

$$\widehat{\mathbf{m}}_t = \frac{\beta_1\mathbf{m}_{t-1} + (1-\beta_1)\mathbf{g}_t}{1-\beta_1^t}, \quad \widehat{\mathbf{v}}_t = \frac{\beta_2\mathbf{v}_{t-1} + (1-\beta_2)\mathbf{g}_t^2}{1-\beta_2^t}, \qquad \mathbf{D}_t = \mathbf{diag}\left(\sqrt{\widehat{\mathbf{v}}_t} + \epsilon\mathbf{1}\right),$$

where $\beta_1$ and $\beta_2$ are momentum coefficients and $\epsilon$ is a numerical stability constant.

**LION-$\mathcal{K}$.** Given a convex function $\mathcal{K}$, the update vector $\mathbf{u}_t$ is a momentum-filtered step that is preconditioned using a subgradient, i.e.

$$\mathbf{m}_t = \beta_2\mathbf{m}_{t-1} - (1-\beta_2)\mathbf{g}_t, \quad \widetilde{\mathbf{m}}_t = \beta_1\mathbf{m}_{t-1} - (1-\beta_1)\mathbf{g}_t, \quad \mathbf{u}_t = -\nabla\mathcal{K}(\widetilde{\mathbf{m}}_t),$$

where $\beta_1$ and $\beta_2$ are momentum coefficients and $\nabla\mathcal{K}$ is a subgradient of $\mathcal{K}$. Examples include LION when $\mathcal{K} = \|\cdot\|_1$ and MUON when $\mathcal{K} = \|\cdot\|_{\mathrm{tr}}$, where $\|\cdot\|_{\mathrm{tr}}$ denotes the nuclear norm when the parameters are treated as a matrix.

## 2.2 IMPLICIT REGULARIZATION EFFECTS OF WEIGHT DECAY

In general, the application of decoupled weight decay imposes a certain regularization or constraint effect on the objective function, where the specific effect depends on the choice of $\mathbf{u}_t$. For example, SGD with decoupled weight decay is exactly SGD on an $\ell_2$-regularized objective. To see the equivalence, let $f : \mathbb{R}^d \to \mathbb{R}$ be differentiable and consider the regularized variant $\widehat{f}(\mathbf{x}) := f(\mathbf{x}) + \frac{\lambda}{2}\|\mathbf{x}\|_2^2$. A single SGD step on $\widehat{f}$ with learning rate $\eta_t > 0$ yields the update

$$\mathbf{x}_{t+1} = \mathbf{x}_t - \eta_t(\nabla f(\mathbf{x}_t) + \lambda\mathbf{x}_t) = (1 - \eta_t\lambda)\mathbf{x}_t - \eta_t\nabla f(\mathbf{x}_t),$$

which is precisely the decoupled weight decay update given by (1).

Given a convex function $\mathcal{K}$ with subgradient $\nabla\mathcal{K}$ and convex conjugate $\mathcal{K}^*$, suppose the iterates of LION-$\mathcal{K}$ converge to a fixed point $(\mathbf{x}^\star, \mathbf{m}^\star, \widetilde{\mathbf{m}}^\star)$. Then the moment estimators stabilize so that $\mathbf{m}^\star = \widetilde{\mathbf{m}}^\star = -\nabla f(\mathbf{x}^\star)$, and the fixed-point condition yields $-\nabla\mathcal{K}(-\nabla f(\mathbf{x}^\star)) + \lambda\mathbf{x}^\star = \mathbf{0}$. Rearranging and using the identity $(\nabla\mathcal{K})^{-1} = \nabla\mathcal{K}^*$, we obtain $\nabla f(\mathbf{x}^\star) + \nabla\mathcal{K}^*(\lambda\mathbf{x}^\star) = \mathbf{0}$, where the left-hand side is the gradient of the function

$$\widehat{f}(\mathbf{x}) := f(\mathbf{x}) + \frac{1}{\lambda}\mathcal{K}^*(\lambda\mathbf{x}).$$

This suggests that LION-$\mathcal{K}$ optimizes the regularized objective $\widehat{f}$, an observation made by Chen et al. (2024). In the special cases of LION and MUON, $\mathcal{K}^*$ is the 0-$\infty$ indicator function of a dual norm ball, corresponding to the constrained optimization problems

$$\min_{\mathbf{x}\in\mathbb{R}^d} f(\mathbf{x}) \quad \text{s.t.} \quad \|\mathbf{x}\|_\infty \leq \frac{1}{\lambda} \qquad \text{and} \qquad \min_{\mathbf{X}\in\mathbb{R}^{n\times m}} f(\mathbf{X}) \quad \text{s.t.} \quad \|\mathbf{X}\|_{\mathrm{op}} \leq \frac{1}{\lambda},$$

respectively, where $\|\cdot\|_{\mathrm{op}}$ is the spectral norm when the parameters are treated as a matrix.

A similar analysis for ADAMW suggests that it solves the box-constrained problem of minimizing $f(\mathbf{x})$ such that $\|\mathbf{x}\|_\infty \leq \frac{1}{\lambda}$, but convergence cannot be established due to the lack of a Lyapunov function. For more discussion, see Appendix C and Xie & Li (2024).

While ADAMW and LION-$\mathcal{K}$ are practically strong, they implicitly optimize a regularized surrogate that is dependent on the weight decay coefficient $\lambda$. This motivates the development of a mechanism that maintains the beneficial effects of decoupled weight decay (e.g. regularization, training acceleration) while optimizing the original objective.

## 3 CAUTIOUS WEIGHT DECAY

Cautious weight decay (CWD) modifies the update rule (1) as

$$\mathbf{x}_{t+1} = \mathbf{x}_t - \eta_t(\mathbf{u}_t + \lambda\mathbb{I}(\mathbf{u}_t \odot \mathbf{x}_t \geq \mathbf{0}) \odot \mathbf{x}_t),$$

where $\odot$ denotes entrywise multiplication.[1] As a one-line modification, cautious weight decay is implementation-trivial and universally compatible with gradient-based optimization algorithms. Theoretically, cautious weight decay also exhibits the following behavior.

• **Unbiased optimization**, in the sense that every accumulation point $\mathbf{x}^\star$ of the trajectory satisfies $\nabla f(\mathbf{x}^\star) = \mathbf{0}$ under the same convergence conditions required of the base optimizer without weight decay. In over-parameterized deep models, the set of stationary points is typically a union of connected submanifolds rather than isolated points. Consequently, the $\omega$-limit set of the trajectory is contained in some stationary manifold, and the iterates eventually remain arbitrarily close to it.

• **Sliding mode dynamics** within the stationary manifold, where cautious weight decay allows the trajectory to traverse along the manifold until it cannot decrease the parameter magnitudes in every coordinate. In other words, cautious weight decay steers the trajectory towards a local Pareto front of the stationary manifold under the ordering that prioritizes smaller parameter magnitudes.

### 3.1 CONVERGENCE TO THE STATIONARY MANIFOLD

We construct Lyapunov functions for the continuous-time limits of several standard optimizers equipped with cautious weight decay. A *Lyapunov function* is a lower bounded function with non-positive derivative that is used to certify the stability of systems of differential equations.

Consider the continuous-time dynamics of SGD with cautious weight decay

$$\dot{\mathbf{x}}_t = -\nabla f(\mathbf{x}_t) - \lambda \mathbb{I}(\nabla f(\mathbf{x}_t)\mathbf{x}_t \geq \mathbf{0})\mathbf{x}_t.$$

This ODE has the Lyapunov function $\mathcal{H}(\mathbf{x}) = f(\mathbf{x})$, since $\mathcal{H}$ is lower bounded and

$$\frac{\mathrm{d}\mathcal{H}}{\mathrm{d}t} = \langle \nabla f(\mathbf{x}_t), -\nabla f(\mathbf{x}_t) - \lambda \mathbb{I}(\nabla f(\mathbf{x}_t)\mathbf{x}_t \geq \mathbf{0})\mathbf{x}_t \rangle = -\|\nabla f(\mathbf{x}_t)\|_2^2 - \lambda \left\|(\nabla f(\mathbf{x}_t)\mathbf{x}_t)^+\right\|_1 \leq 0,$$

where $(\cdot)^+ := \max(0, \cdot)$. LaSalle's invariance principle (LaSalle, 1960) states that the accumulation points of any trajectory lie within the union of trajectories $\mathbf{z}_t$ that satisfy $\frac{\mathrm{d}}{\mathrm{d}t}\mathcal{H}(\mathbf{z}_t) = 0$ for all $t \geq 0$. Consequently, we conclude that SGD with cautious weight decay produces trajectories that approach the stationary set $\{\mathbf{x} \mid \nabla f(\mathbf{x}) = \mathbf{0}\}$ of the original loss. This holds because cautious weight decay is applied only in a secondary fashion and is automatically deactivated whenever it conflicts with the main objective, thereby ensuring that the loss landscape remains unbiased.

Beyond the simple case of SGD, the same Lyapunov-type argument can be extended to more sophisticated algorithms such as SGDM, LION-$\mathcal{K}$, and ADAM. In each case, cautious weight decay still minimizes the original objective without introducing explicit bias, but a key difficulty lies in constructing appropriate Lyapunov functions. Table 1 summarizes the Lyapunov functions of several major optimizers with cautious weight decay, and detailed derivations are provided in Appendix D. By applying LaSalle's invariance principle, we can show that the momentum-based algorithms in Table 1 converge to the stationary set of the original objective, together with vanishing momentum:

$$\{(\mathbf{x}, \mathbf{m}) \mid \nabla f(\mathbf{x}) = \mathbf{0}, \ \mathbf{m} = \mathbf{0}\}.$$

### 3.2 SLIDING MODE DYNAMICS

Although both standard optimization (with no weight decay) and cautious weight decay are unbiased with respect to the original objective, their behaviors diverge within the stationary manifold. In the former, the dynamics halt as the momentum $\mathbf{m}$ decays to zero, while, in contrast, the cautious weight decay dynamics induce a *sliding mode*, continuing to move along the manifold while reducing the parameter magnitudes as much as possible. Consequently, the algorithm converges to a subset of the stationary manifold where further simultaneous reduction of all coordinates of $\mathbf{x}$ is no longer possible. Equivalently, it converges to a locally Pareto-optimal stationary point under a preference for smaller parameter magnitudes.

To provide mathematical background, consider a possibly time-varying discontinuous ODE

$$\dot{\mathbf{z}}_t = f_t(\mathbf{z}_t), \quad \mathbf{z}_t \in \mathbb{R}^d.$$

---

[1]Throughout the paper, when it is clear from context, we also drop $\odot$ and write $\mathbf{v} \odot \mathbf{x} = \mathbf{v}\mathbf{x}$ for simplicity.

Table 1: Comparison of the continuous-time dynamics of different optimizers. SGDM represents SGD with momentum. LION-$\mathcal{K}$ includes LION ($\mathcal{K} = \|\cdot\|_1$) and MUON ($\mathcal{K} = \|\cdot\|_{\mathrm{tr}}$) as special cases. $f : \mathbb{R}^d \to \mathbb{R}$ is assumed to be differentiable and lower bounded by $f^\star$.

| Optimizer | Continuous-time dynamics | Lyapunov function |
|---|---|---|
| SGD + CWD | $\dot{\mathbf{x}}_t = -\nabla f(\mathbf{x}_t) - \lambda \mathbb{I}(\nabla f(\mathbf{x}_t)\mathbf{x}_t \geq \mathbf{0})\mathbf{x}_t$ | $\mathcal{H}(\mathbf{x}) = f(\mathbf{x})$ |
| SGDM + CWD | $\dot{\mathbf{x}}_t = -\mathbf{m}_t - \lambda \mathbb{I}(\mathbf{m}_t\mathbf{x}_t \geq \mathbf{0})\mathbf{x}_t$ 
 $\dot{\mathbf{m}}_t = \beta(\nabla f(\mathbf{x}_t) - \mathbf{m}_t)$ | $\mathcal{H}(\mathbf{x}, \mathbf{m}) = \beta f(\mathbf{x}) + \frac{1}{2}\|\mathbf{m}\|_2^2 + \lambda \left\|(\mathbf{mx})^+\right\|_1$ |
| LION-$\mathcal{K}$ + CWD | $\dot{\mathbf{x}}_t = \nabla\mathcal{K}(\mathbf{m}_t) - \lambda \mathbb{I}(\mathbf{m}_t\mathbf{x}_t \leq \mathbf{0})\mathbf{x}_t$ 
 $\dot{\mathbf{m}}_t = -\alpha\nabla f(\mathbf{x}_t) - \gamma\mathbf{m}_t$ | $\mathcal{H}(\mathbf{x}, \mathbf{m}) = \alpha f(\mathbf{x}) + \mathcal{K}(\mathbf{m}) + \lambda \left\|(-\mathbf{mx})^+\right\|_1$ |
| ADAM + CWD | $\dot{\mathbf{x}}_t = -\dfrac{\alpha_t\mathbf{m}_t}{\mathbf{h}_t} - \lambda \mathbb{I}(\mathbf{m}_t\mathbf{x}_t \geq \mathbf{0})\mathbf{x}_t$ 
 $\dot{\mathbf{m}}_t = \alpha(\nabla f(\mathbf{x}_t) - \mathbf{m}_t)$ 
 $\dot{\mathbf{v}}_t = \gamma(\nabla f(\mathbf{x}_t)^2 - \mathbf{v}_t)$ | $\mathcal{H}_t(\mathbf{x}, \mathbf{m}, \mathbf{h}) = \alpha f(\mathbf{x}) + \left\|\dfrac{\alpha_t\mathbf{m}^2}{2\mathbf{h}}\right\|_1 + \lambda \left\|(\mathbf{mx})^+\right\|_1$ |

*Notation.* We drop $\odot$ for simplicity. $\alpha_t := (1 - \exp(-\alpha t))^{-1}$, $\gamma_t := (1 - \exp(-\gamma t))^{-1}$, $\mathbf{h}_t := \sqrt{\gamma_t\mathbf{v}_t} + \epsilon\mathbf{1}$.

Due to the discontinuity of $f_t$, the solution may not be well defined in the classical or Carathéodory sense, especially across switching surfaces. We therefore interpret solutions in the Filippov sense (Filippov, 1988), where a discontinuous ODE is formally a *differential inclusion* that specifies that $\dot{\mathbf{z}}_t$ belongs to the closed convex envelope of the discontinuous vector field, i.e.

$$\dot{\mathbf{z}}_t \in \mathcal{F}[f_t](\mathbf{z}_t) := \bigcap_{\delta > 0} \bigcap_{\mu(S)=0} \overline{\mathrm{co}}(f_t(\mathbb{B}(\mathbf{z}_t, \delta) \setminus S)),$$

where $\mu$ denotes the Lebesgue measure, $\mathbb{B}(\mathbf{z}, \delta)$ is the $\delta$-ball centered at $\mathbf{z}$, and $\overline{\mathrm{co}}$ denotes the closed convex envelope. This construction captures all possible limiting directions of the vector field near discontinuities, ensuring well-defined dynamics even when $f_t$ is not continuous. The key idea is that the values of $\dot{\mathbf{z}}_t$ must be determined by the behavior of $f_t$ in a neighborhood around $\mathbf{z}_t$, rather than at the point itself. The inclusion, therefore, defines a range of admissible velocities consistent with the nearby values of the vector field.

In particular, whenever $f_t$ contains coordinatewise indicators such as $\mathbb{I}(g(\mathbf{z}_t) \geq 0)$, the Filippov set replaces them by *selectors* $\mathbf{s}_t \in [0, 1]^d$ on the switching set $\{[g(\mathbf{z}_t)]_i = 0\}$:

$$[\mathbf{s}_t]_i \in \begin{cases} \{1\} & [g(\mathbf{z}_t)]_i > 0, \\ \{0\} & [g(\mathbf{z}_t)]_i < 0, \\ [0, 1] & [g(\mathbf{z}_t)]_i = 0. \end{cases}$$

Recalling the Lyapunov analysis in Section 3.1, the continuous-time dynamics of standard optimizers with cautious weight decay converge to the stationary manifold $\mathbb{M} := \{\mathbf{x} \mid \nabla f(\mathbf{x}) = \mathbf{0}\}$, with the momentum $\mathbf{m}_t$ also decaying to $\mathbf{0}$ for momentum-based methods. Consequently, once the trajectory enters the stationary manifold, the residual dynamics reduce to

$$\dot{\mathbf{x}}_t = -\lambda\mathbf{s}_t \odot \mathbf{x}_t, \qquad \mathbf{s}_t \in [0, 1]^d. \tag{2}$$

Moreover, since the Lyapunov function confines the dynamics to the stationary set, the selectors $\mathbf{s}_t$ must be chosen such that the trajectory remains within the manifold. Differentiating the stationarity condition yields

$$\frac{\mathrm{d}}{\mathrm{d}t}\nabla f(\mathbf{x}_t) = -\lambda\nabla^2 f(\mathbf{x}_t)(\mathbf{s}_t \odot \mathbf{x}_t) = \mathbf{0}, \qquad \mathbf{s}_t \in [0, 1]^d.$$

This relation allows us to solve for admissible choices of $\mathbf{s}_t$ that guarantee invariance of the manifold. In general, the solution for $\mathbf{s}_t$ need not be unique, and the actual value realized in practice may be implicitly determined by the discretization scheme employed.

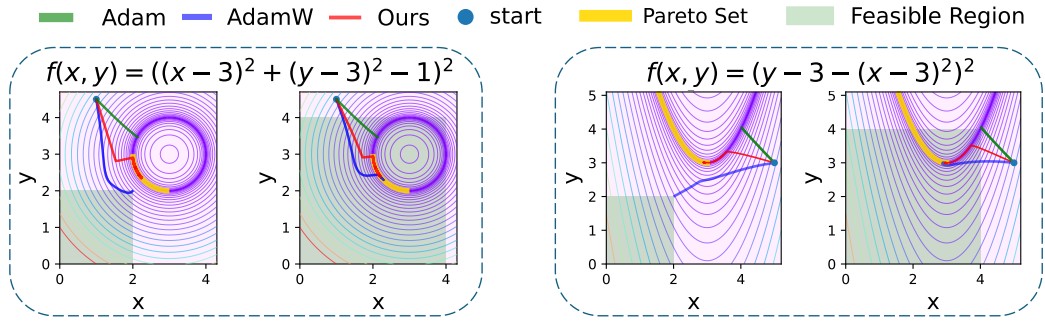

Figure 3: Toy objectives and trajectories. **Left**: $f(x,y) = ((y-3)^2 + (x-3)^2 - 1)^2$. **Right**: $f(x,y) = (y - 3 - (x-3)^2)^2$. We compare ADAM, ADAMW, and ADAM + CWD; ADAMW and CWD use the same weight decay $\lambda$, and all other hyperparameters $(\eta, \beta_1, \beta_2, \epsilon)$ are identical. For both objectives, ADAM converges to a generic point on the minimizer manifold, whereas ADAMW converges to a solution of the box-constrained problem $\min_{x,y} f(x,y)$ subject to $\max\{x,y\} \leq \frac{1}{\lambda}$. In contrast, ADAM + CWD converges to the Pareto front of the minimizer manifold.

Effectively, cautious weight decay decreases parameter magnitudes along each coordinate while staying within the stationary manifold, pushing $\mathbf{x}$ toward the *local Pareto front* of the manifold

$$\mathbb{P} := \{\mathbf{x} \in \mathbb{M} \mid \exists \delta > 0 \, \forall \mathbf{y} \in (\mathbb{B}(\mathbf{x}, \delta) \cap \mathbb{M}) \setminus \{\mathbf{x}\}, |\mathbf{y}| \not\leq |\mathbf{x}|\},$$

where the tangent space no longer allows a nonzero $\mathbf{s}_t$ in (2). In other words, a stationary point is locally Pareto-optimal if it has a neighborhood in the stationary manifold that contains no other point with a smaller or equal magnitude in every coordinate.

This argument shows that cautious weight decay dynamics converge to $\mathbb{P}$. Since $\mathbb{P}$ may not be a singleton, the exact limit point depends intricately on initialization and the discretization of the continuous-time dynamics. Figure 3 illustrates this behavior on two toy problems.

### 3.3 DISCRETE-TIME ANALYSIS

Leveraging the Lyapunov functions in Table 1, we can extend our continuous-time analysis to obtain convergence guarantees for the discrete-time dynamics of various optimizers with cautious weight decay. As a concrete example, we provide in Appendix E an explicit convergence rate for discrete-time ADAM with cautious weight decay.

## 4 EXPERIMENTS

**Overview.** We evaluate CWD against three standard optimizers—ADAMW, LION, and MUON—on autoregressive language modeling and ImageNet classification. For Transformer models with similar architecture to Gemma (Kamath et al., 2025) with 338M, 986M, and 2B parameters in the Simply (Liang et al., 2025) codebase, we follow the Chinchilla compute-optimal scaling rule—20 tokens per parameter (TPP) (Hoffmann et al., 2022) and train on C4 (Raffel et al., 2020). For each size, we grid-search batch size, learning rate, weight decay, warmup ratio, and optimizer-specific hyperparameters for the baselines (ADAMW, LION, MUON); *we then reuse the selected baseline settings for CWD without retuning* (details in Appendix F). Under matched settings, CWD lowers final validation loss and improves zero-shot accuracy. On the OLMo codebase (OLMo et al., 2025), we further study an over-training regime—OLMo-1B trained on 100B tokens (100 TPP) from Dolma (Soldaini et al., 2024). Under matched settings, CWD lowers final validation loss and improves zero-shot accuracy (Table 4). We also observe similar gains on ImageNet (Deng et al., 2009) across ViT (Dosovitskiy et al., 2021) and ResNet (He et al., 2016).

**Ablations of weight decay.** Figure 1 sweeps the weight–decay coefficient $\lambda$ for a 338M model on C4: $\lambda \in [0, 0.4]$ for MUON and ADAMW, and $\lambda \in [0, 3.0]$ for LION. Two patterns are consistent across runs: (i) at a fixed $\lambda$, CWD attains a lower final loss than the corresponding baseline with decoupled weight decay; (ii) the minimizing value $\lambda^\star$ is essentially unchanged when replacing the baseline with CWD. In practice, one can swap in CWD at an already tuned $\lambda$ and obtain improvements without additional sweeps.

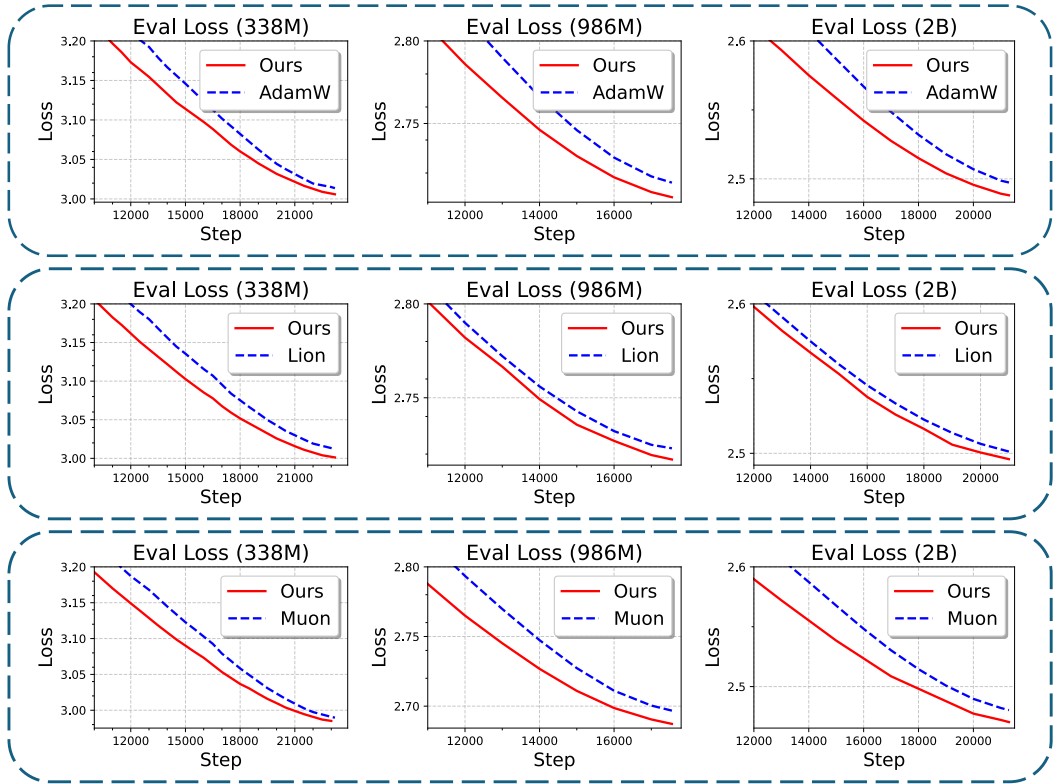

Figure 4: **Evaluation loss across scales.** $3\times3$ grid for 338M, 986M, and 2B Transformer models trained with ADAMW, LION, and MUON on C4 dataset. *All panels show a zoom into the final ~40% of training steps* to highlight late-stage behavior. Baseline curves (dashed blue) use standard weight decay with tuned hyperparameters (learning rate schedule, $\beta$'s, weight decay, etc.; see Appendix F). Our method (solid red) follows Algorithm 1 and reuses the baseline hyperparameters without additional tuning. Full (non-zoomed) curves are in Figures 8, 9 and 10 in Appendix G.

Table 2: Ablation study of selective weight decay strategies on OLMo-1B (100B tokens). We compare our momentum-based selection against alternative masking approaches. **Baseline**: standard weight decay ($\lambda$ tuned). **Ours**: update-based mask $\mathbb{I}(\mathbf{ux} \geq 0)$ using baseline's $\lambda$ without retuning. **Random**: time-varying Bernoulli mask matching our method's sparsity ratio (see Figure 7 in Appendix G). **Gradient**: uses $\mathbb{I}(\mathbf{gx} \geq 0)$ instead. **No WD**: $\lambda = 0$. Lower validation loss is better.

| | Weight Decay Active | | Ablated Masks | | Disabled |
|---|---|---|---|---|---|
| **Optimizer** | Baseline | **Ours** | Random | Gradient | No WD |
| ADAMW | 2.65 | **2.56** | 2.82 | 2.75 | 2.70 |
| MUON | 2.51 | **2.42** | 2.73 | 2.74 | 2.62 |

**Ablations on masking.** Table 2 tests whether the benefits arise from the *amount* of decay applied or from CWD's *structure*. Replacing our mask with a time-matched Bernoulli "random mask" substantially degrades performance (e.g., $2.56 \rightarrow 2.82$ for ADAMW, $2.42 \rightarrow 2.73$ for MUON), showing that simply reducing the frequency of decay is insufficient. Substituting the indicator with the gradient-based $\mathbb{I}(\mathbf{gx} \geq \mathbf{0})$ also underperforms. Finally, $\lambda = 0$ remains worse than tuned decay, illustrating that explicit regularization is helpful and CWD leverages it more effectively. We further verify the difference between CWD and SPD (Tian et al., 2024) with additional instruction-following fine-tuning experiments. We fine-tune on the Alpaca GPT-4 dataset (Peng et al., 2023), which contains 52,000 instruction–response pairs generated by GPT-4 (OpenAI, 2023), and evaluate on three benchmarks: MMLU (Hendrycks et al., 2021), comprising 57 tasks and 14,079 questions covering a broad range of world knowledge; AGIEval (Zhong et al., 2024), a human-centric benchmark with 9,316 instances targeting general reasoning and problem-solving skills; and WinoGrande (Sakaguchi et al., 2021), a large-scale commonsense reasoning dataset with 44,000 instances. We consider two base models,

Table 3: ImageNet validation accuracy (%) across architectures and optimizers. All models train for 300 epochs with standard augmentation. **Base**: optimizer with tuned weight decay. **Ours**: cautious weight decay using the same coefficient as baseline (no retuning).

| Model | Params | ADAMW | | LION | | MUON | |
|---|---|---|---|---|---|---|---|
| | | Base | Ours | Base | Ours | Base | Ours |
| ViT-S/16 | 22.05M | 78.84 | **79.45** | 79.29 | **79.82** | 79.35 | **79.91** |
| ResNet-50 | 25.56M | 76.30 | **76.68** | 76.41 | **76.75** | 76.47 | **76.83** |
| ViT-B/16 | 86.57M | 80.15 | **80.71** | 80.76 | **80.92** | 80.83 | **81.04** |

| Optimizer | Hellaswag ↑ acc_norm | ARC-Easy ↑ acc_norm | ARC-C ↑ acc_norm | PIQA ↑ acc_norm | MMLU ↑ acc | ComQA ↑ acc |
|---|---|---|---|---|---|---|
| ADAMW | 0.38 | 0.50 | 0.25 | 0.67 | 0.23 | 0.29 |
| ADAMW+CWD | **0.40** | **0.53** | **0.27** | **0.69** | **0.25** | **0.31** |
| MUON | 0.39 | **0.51** | 0.26 | 0.68 | 0.24 | 0.30 |
| MUON+CWD | **0.41** | **0.51** | **0.28** | **0.71** | **0.26** | **0.33** |

Table 4: Downstream accuracy across diverse reasoning benchmarks. All runs use the OLMo codebase with 1B-parameter models trained for 100B tokens under an over-training regime. Here ARC-C=ARC-Challenge and ComQA=CommonsenseQA. Figure 5 shows the corresponding loss curves.

TinyLlama (Zhang et al., 2024) and Mistral-7B (Jiang et al., 2023), and compare LoRA (Hu et al., 2022), SPD (Tian et al., 2024), a layerwise "inner-product" variant of CWD using $\mathbb{I}(\langle \mathbf{u}, \mathbf{x} \rangle \geq 0)$, and our proposed "elementwise" CWD. Table 5 reports accuracy on MMLU, AGIEval, and WinoGrande for TinyLlama and Mistral-7B.

**Training dynamics.** On 1B models trained for 100B tokens, we observe that CWD tends to improve the loss trajectory relative to tuned ADAMW and MUON, rather than only the final value (Figure 5). A similar pattern appears at 986M: Figure 11 in Appendix G shows evaluation/training loss and RMS parameter norm over time. CWD generally achieves lower loss while ending with an intermediate norm. In contrast, removing decay entirely ($\lambda = 0$) descends faster mid-training but plateaus earlier, finishing at higher loss and the largest norm; tuned ADAMW with $\lambda > 0$ yields the smallest norm. Overall, these results suggest that the gains come from a more selective application of regularization rather than from disabling it.

**CWD outperforms standard decay across optimizers and scales.** Under the common setup across 338M, 986M, and 2B parameters, CWD consistently lowers eval loss for ADAMW, LION, and MUON (see Figure 4 and Figures 8–10 in Appendix G) and increases downstream accuracy (Table 4).

**CWD yields lower gradient norms than standard decay.** Across model sizes, CWD produces lower RMS-normalized gradient norms than the corresponding baselines (see Figure 12 in Appendix G). This coincides with the lower end-of-training loss in Figure 5 and the accuracy gains in Table 4.

| Model | Method | MMLU (5-shot) ↑ | AGIEval (3-shot) ↑ | WinoGrande (5-shot) ↑ |
|---|---|---|---|---|
| TinyLlama | LoRA (baseline) | 25.81 ± 0.07 | 19.82 ± 0.11 | 61.33 ± 0.09 |
| TinyLlama | SPD | 26.14 ± 0.08 | **20.21 ± 0.10** | 61.92 ± 0.08 |
| TinyLlama | Inner-product CWD | 26.02 ± 0.08 | 19.80 ± 0.10 | 61.70 ± 0.09 |
| TinyLlama | Elementwise CWD | **26.42 ± 0.09** | 20.12 ± 0.09 | **62.18 ± 0.08** |
| Mistral-7B | LoRA (baseline) | 61.78 ± 0.09 | 27.56 ± 0.07 | 78.85 ± 0.11 |
| Mistral-7B | SPD | 62.05 ± 0.08 | 27.98 ± 0.06 | 78.81 ± 0.10 |
| Mistral-7B | Inner-product CWD | 61.76 ± 0.09 | 27.90 ± 0.07 | 78.83 ± 0.10 |
| Mistral-7B | Elementwise CWD | **62.13 ± 0.07** | **28.31 ± 0.06** | **78.92 ± 0.09** |

Table 5: Accuracy on MMLU, AGIEval, and WinoGrande for TinyLlama and Mistral-7B fine-tuned on Alpaca GPT-4 using LoRA, SPD, and two variants of CWD. Both SPD and CWD improve over the LoRA baseline, and the proposed *elementwise* CWD matches or outperforms SPD and the inner-product variant on most benchmarks.

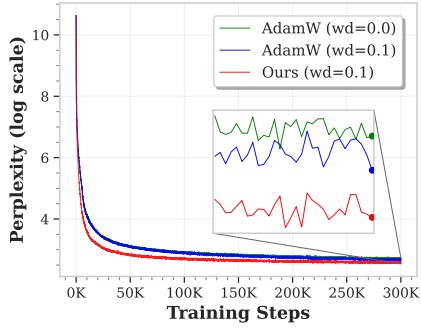 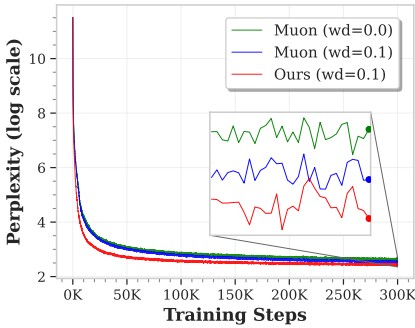

Figure 5: Training loss of OLMo 1B on 100B tokens. **Left**: ADAMW. **Right**: MUON.

**Scaling with model size.** We measured final validation loss for ADAMW and CWD ("Ours") at 111M, 338M, 986M, and 2B parameters on the same dataset (Figure 6). At every scale, CWD attains a lower final validation loss than ADAMW, and the gap remains stable or even widens with model size, indicating that the advantages of cautious weight decay persist into the large-model regime.

Figure 6: Final validation loss versus model size for ADAMW and ADAM + CWD ("Ours") on the same pretraining dataset, at 111M, 338M, 986M, and 2B parameters. Across all scales, ADAM + CWD achieves consistently lower final validation loss than ADAMW, and the gap remains stable or slightly increases with model size, suggesting that the benefits of cautious weight decay persist in the large-model regime.

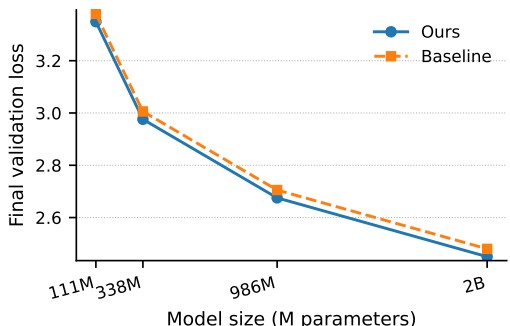

| Model Size | Optimizer | Final Validation Loss |
|---|---|---|
| 338M | ADAMW (baseline) | 3.0136 |
| 338M | ADAM + CWD | 3.0059 |
| 338M | ADAMC | 3.0087 |
| 338M | ADAMC + CWD | **2.9915** |

Table 6: Final validation loss for Gemma-based models with different optimizers at 338M parameters. Adding cautious weight decay improves both ADAMW and ADAMC, with **ADAMC + CWD** achieving the lowest loss.

| Model Size | Optimizer | Scheduler | Final Validation Loss |
|---|---|---|---|
| 338M | ADAMW (baseline) | Cosine | 3.0136 |
| 338M | ADAMW (baseline) | WSD | 3.0101 |
| 338M | ADAM + CWD | Cosine | **3.0059** |
| 338M | ADAM + CWD | WSD | **3.0014** |

Table 7: Final validation loss for a 338M model under different optimizer–scheduler combinations. ADAM + CWD improves over the ADAMW baseline for both cosine and WSD schedules, with the best performance obtained by ADAM + CWD with WSD.

**Compound improvement with other techniques.** We also observe *compounding gains* when combining cautious weight decay with other optimizer techniques on Gemma-based model architectures. In particular, we compare ADAMW, ADAMC (Defazio, 2025), and their variants with CWD at 338M parameters (Table 6).

**Robustness to learning-rate schedules.** We observe that `CWD` improves performance for both cosine scheduling and the Warmup–Stable–Decay (WSD) schedule, which uses a 10% warmup, a long stable phase, and a 20% final decay (Table 7).

## 5 RELATED WORK

**Weight decay.** Weight decay originated as an $\ell_2$ penalty for ill-posed problems and ridge regression (Tikhonov, 1963; Hoerl & Kennard, 1970) and was introduced to neural networks as a generalization tool to mitigate overfitting (Hanson & Pratt, 1988; Weigend et al., 1990; Krogh & Hertz, 1991). Loshchilov & Hutter (2019) showed that, for adaptive methods, weight decay and $\ell_2$ regularization are not equivalent, motivating the decoupled formulation in ADAMW; subsequent work established decoupled decay as a standard feature of modern optimizers (Chen et al., 2023; 2024; Liu et al., 2025). Recent analyses suggest that in contemporary networks, weight decay functions more as a training accelerator and stabilizer than as explicit regularization (Krizhevsky et al., 2017; Hoffmann et al., 2022; Pan & Cao, 2023; D'Angelo et al., 2024). Interactions with early-stage training (Bjorck et al., 2021), normalization layers, and learning rate schedules (Defazio, 2025) have also been clarified, and architectural designs can obviate explicit decay (Loshchilov et al., 2025).

**Weight decay variants.** Various efforts have been made to develop different adaptive variants of weight decay. For example, Xie et al. (2023) found that weight decay can lead to large gradient norms at the final phase of training and proposed Scheduled Weight Decay (SWD) to dynamically adjust weight decay strength based on gradient norms. Kosson et al. (2024) investigate how weight decay affects individual neuron updates, revealing rotational equilibrium states that balance learning across layers and neurons. Ghiasi et al. (2023) introduce adaptive weight decay that automatically tunes the hyperparameter during training based on classification and regularization loss gradients, achieving significant improvements in adversarial robustness. Tian et al. (2024) introduce Selective Projection Decay (SPD) for robust fine-tuning, featuring selective weight decay via a mask that is somewhat similar to `CWD`. However, SPD and `CWD` differ in significant ways, including the intended setting, mechanism, theoretical properties, and empirical performance.

**Masked or conditional updates.** Several works have explored the sign-based conditioning of optimizer updates. Riedmiller & Braun (1993) introduced RPROP, which adjusted step sizes based on current gradient and past gradient sign agreement. Liang et al. (2024a) propose the cautious optimizer, which restricts updates to dimensions where the proposed update and current gradient share the same sign. Wang et al. (2024) apply a similar mask to ADAM to improve robustness in online learning.

**Constrained and bilevel optimization.** Decoupled weight decay can be interpreted through the lens of Frank–Wolfe algorithms for constrained optimization (Frank & Wolfe, 1956; Jaggi, 2013; Sfyraki & Wang, 2025; Pethick et al., 2025). This connection suggests that optimizers with decoupled weight decay implicitly solve constrained optimization problems, which was shown to be the case for LION (Chen et al., 2024; Sfyraki & Wang, 2025; Pethick et al., 2025), ADAMW (Xie & Li, 2024; Bernstein & Newhouse, 2024), and MUON (Chen et al., 2025; Sfyraki & Wang, 2025; Lau et al., 2025). In contrast, optimizers with cautious weight decay perform bilevel optimization, a framework from classical optimization (Solodov, 2007a;b; Sabach & Shtern, 2017) that has been recently explored in machine learning (Gong et al., 2021; Liu et al., 2022; Petrulionyte et al., 2024).

## 6 CONCLUSION

We introduce cautious weight decay and formalize it as a simple, optimizer-agnostic modification of decoupled weight decay that preserves the optimization objective while retaining the practical benefits of weight decay. For standard optimizers (SGD, ADAM, and LION-$\mathcal{K}$), we show the bilevel optimization structure of cautious weight decay and establish convergence guarantees in both continuous- and discrete-time regimes. Across diverse tasks and benchmarks, cautious weight decay consistently improves training dynamics compared to no decay and traditional decoupled decay, yielding faster loss reduction and more stable trajectories without changes to hyperparameters or model architectures. Our results indicate that cautious weight decay is a theoretically principled and empirically effective technique that retains the benefits of weight decay while addressing its fundamental limitations.

## ACKNOWLEDGMENTS

This work was supported in part by the Institute for Foundations of Machine Learning (IFML) and the Office of Naval Research (ONR) under Grant No. N00014-25-1-2354.

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

## A    NOTATION AND DEFINITIONS

$\mathbb{N} := \{1, 2, 3, \dots\}$ denotes the natural numbers. For $n \in \mathbb{N}$, $[n]$ denotes the set $\{1, 2, \dots, n\}$. Vectors are denoted in lowercase boldface, and matrices are denoted in capital boldface. $\mathbf{0}$ and $\mathbf{1}$ denote the all-zeros and all-ones tensors of appropriate dimension, respectively. Scalar operations and functions, e.g. multiplication, division, and square roots, are understood to be performed entrywise when applied to vectors. We also use $\odot$ to explicitly denote the entrywise product. $x^+$ denotes the positive part of $x$, i.e.

$$x^+ := \max(0, x) = \begin{cases} x & \text{if } x > 0 \\ 0 & \text{otherwise} \end{cases}.$$

$\|\cdot\|_p$ denotes the $\ell_p$ norm for $p \in [1, \infty]$. $\langle \cdot, \cdot \rangle$ denotes the standard inner product on $\mathbb{R}^d$. $[\mathbf{x}]_i$ denotes the $i^{\text{th}}$ entry of a vector $\mathbf{x}$. $\mathbf{diag}(\mathbf{x})$ denotes the diagonal matrix with diagonal entries given by $\mathbf{x}$. $\mathbb{I}(\mathbf{x} \geq \mathbf{0})$ denotes the indicator tensor that is 1 in a coordinate if $\mathbf{x}$ is nonnegative in that coordinate and 0 otherwise. If $\mathcal{K} : \mathbb{R}^d \to \mathbb{R}$ is convex, we let $\partial \mathcal{K}(\mathbf{x})$ denote the set of subgradients of $\mathcal{K}$ at $\mathbf{x}$ and overload $\nabla \mathcal{K}(\mathbf{x})$ to denote an element of $\partial \mathcal{K}(\mathbf{x})$.

**Definition 1** (L-smoothness). *A function $f : \mathbb{R}^d \to \mathbb{R}$ is $L$-smooth if it is differentiable and*

$$\|\nabla f(\mathbf{y}) - \nabla f(\mathbf{x})\|_2 \leq L \|\mathbf{y} - \mathbf{x}\|_2 \quad \text{for all } \mathbf{x}, \mathbf{y} \in \mathbb{R}^d.$$

*If $f$ is L-smooth, then*

$$f(\mathbf{y}) \leq f(\mathbf{x}) + \langle \nabla f(\mathbf{x}), \mathbf{y} - \mathbf{x} \rangle + \frac{L}{2} \|\mathbf{y} - \mathbf{x}\|_2^2 \quad \text{for all } \mathbf{x}, \mathbf{y} \in \mathbb{R}^d.$$

**Definition 2** (Coerciveness). *A function $f : \mathbb{R}^d \to \mathbb{R}$ is coercive if $f(\mathbf{x}) \to \infty$ as $\|\mathbf{x}\| \to \infty$.*

## B    PSEUDOCODE OF OPTIMIZERS WITH CWD

### B.1    SGD WITH MOMENTUM

---
**Algorithm 2** SGD with momentum   and cautious weight decay

---
1: **given** learning rates $\{\eta_t\}_{t \in \mathbb{N}} \subset \mathbb{R}_{>0}$, momentum coefficient $\beta \in [0, 1)$, weight decay coefficient $\lambda > 0$
2: **initialize** time step $t \leftarrow 1$, parameters $\mathbf{x}_1 \in \mathbb{R}^d$, first moment $\mathbf{m}_0 \leftarrow \mathbf{0}$
3: **repeat**
4:     $\mathbf{g}_t \leftarrow \text{StochasticGradient}(\mathbf{x}_t)$
5:     $\mathbf{m}_t \leftarrow \beta \mathbf{m}_{t-1} + (1 - \beta)\mathbf{g}_t$
6:     $\mathbf{x}_{t+1} \leftarrow \mathbf{x}_t - \eta_t \left( \mathbf{m}_t + \lambda \mathbb{I}(\mathbf{m}_t \mathbf{x}_t \geq \mathbf{0})\mathbf{x}_t \right)$                    ▷ entrywise multiplication
7:     $t \leftarrow t + 1$
8: **until** stopping criterion is met
9: **return** optimized parameters $\mathbf{x}_t$

---

### B.2    LION-$\mathcal{K}$

---
**Algorithm 3** LION-$\mathcal{K}$   with cautious weight decay

---
1: **given** learning rates $\{\eta_t\}_{t \in \mathbb{N}} \subset \mathbb{R}_{>0}$, momentum coefficients $\beta_1, \beta_2 \in [0, 1)$, convex $\mathcal{K} : \mathbb{R}^d \to \mathbb{R}$ with subgradient $\nabla \mathcal{K}$, weight decay coefficient $\lambda > 0$
2: **initialize** time step $t \leftarrow 1$, parameters $\mathbf{x}_1 \in \mathbb{R}^d$, first moment $\mathbf{m}_1 \leftarrow \mathbf{0}$
3: **repeat**
4:     $\mathbf{g}_t \leftarrow \text{StochasticGradient}(\mathbf{x}_t)$
5:     $\mathbf{m}_{t+1} \leftarrow \beta_2 \mathbf{m}_t - (1 - \beta_2)\mathbf{g}_t$
6:     $\widetilde{\mathbf{m}}_{t+1} \leftarrow \beta_1 \mathbf{m}_t - (1 - \beta_1)\mathbf{g}_t$
7:     $\mathbf{x}_{t+1} \leftarrow \mathbf{x}_t + \eta_t \left( \nabla \mathcal{K}(\widetilde{\mathbf{m}}_{t+1}) - \lambda \mathbb{I}(\nabla \mathcal{K}(\widetilde{\mathbf{m}}_{t+1})\mathbf{x}_t \leq \mathbf{0})\mathbf{x}_t \right)$        ▷ entrywise multiplication
8:     $t \leftarrow t + 1$
9: **until** stopping criterion is met
10: **return** optimized parameters $\mathbf{x}_t$

---

### B.3 LION

---
**Algorithm 4** LION with cautious weight decay
---

1: **given** learning rates $\{\eta_t\}_{t\in\mathbb{N}} \subset \mathbb{R}_{>0}$, momentum coefficients $\beta_1, \beta_2 \in [0,1)$,
   weight decay coefficient $\lambda > 0$
2: **initialize** time step $t \leftarrow 1$, parameters $\mathbf{x}_1 \in \mathbb{R}^d$, first moment $\mathbf{m}_0 \leftarrow \mathbf{0}$
3: **repeat**
4:     $\mathbf{g}_t \leftarrow \text{StochasticGradient}(\mathbf{x}_t)$
5:     $\widetilde{\mathbf{m}}_t \leftarrow \beta_1 \mathbf{m}_{t-1} + (1-\beta_1)\mathbf{g}_t$
6:     $\mathbf{x}_{t+1} \leftarrow \mathbf{x}_t - \eta_t \left( \text{sgn}(\widetilde{\mathbf{m}}_t) + \lambda \mathbb{I}(\widetilde{\mathbf{m}}_t \mathbf{x}_t \geq \mathbf{0})\mathbf{x}_t \right)$      $\triangleright$ entrywise sgn and multiplication
7:     $\mathbf{m}_t \leftarrow \beta_2 \mathbf{m}_{t-1} + (1-\beta_2)\mathbf{g}_t$
8:     $t \leftarrow t+1$
9: **until** stopping criterion is met
10: **return** optimized parameters $\mathbf{x}_t$

---

### B.4 MUON

---
**Algorithm 5** MUON with cautious weight decay
---

1: **given** learning rates $\{\eta_t\}_{t\in\mathbb{N}} \subset \mathbb{R}_{>0}$, momentum coefficient $\beta \in [0,1)$, weight decay coefficient $\lambda > 0$
2: **initialize** time step $t \leftarrow 1$, parameters $\mathbf{X}_1 \in \mathbb{R}^{n\times m}$, first moment $\mathbf{M}_0 \leftarrow \mathbf{0}$
3: **repeat**
4:     $\mathbf{G}_t \leftarrow \text{StochasticGradient}(\mathbf{X}_t)$
5:     $\mathbf{M}_t \leftarrow \beta \mathbf{M}_{t-1} + \mathbf{G}_t$
6:     $\mathbf{O}_t \leftarrow \text{NewtonSchulz}(\mathbf{M}_t)$      $\triangleright$ approximation of matrix sign
7:     $\mathbf{X}_{t+1} \leftarrow \mathbf{X}_t - \eta_t \left( \mathbf{O}_t + \lambda \mathbb{I}(\mathbf{O}_t \mathbf{X}_t \geq \mathbf{0})\mathbf{X}_t \right)$      $\triangleright$ entrywise matrix multiplication
8:     $t \leftarrow t+1$
9: **until** stopping criterion is met
10: **return** optimized parameters $\mathbf{X}_t$

---

### B.5 ADAM

---
**Algorithm 6** ADAM with cautious weight decay
---

1: **given** learning rates $\{\eta_t\}_{t\in\mathbb{N}} \subset \mathbb{R}_{>0}$, momentum coefficients $0 \leq \beta_1 \leq \beta_2 < 1$, numerical stability
   constant $\epsilon \geq 0$, weight decay coefficient $\lambda > 0$
2: **initialize** time step $t \leftarrow 1$, parameters $\mathbf{x}_1 \in \mathbb{R}^d$, first moment $\mathbf{m}_0 \leftarrow \mathbf{0}$, second moment $\mathbf{v}_0 \leftarrow \mathbf{0}$
3: **repeat**
4:     $\mathbf{g}_t \leftarrow \text{StochasticGradient}(\mathbf{x}_t)$
5:     $\mathbf{m}_t \leftarrow \beta_1 \mathbf{m}_{t-1} + (1-\beta_1)\mathbf{g}_t$
6:     $\mathbf{v}_t \leftarrow \beta_2 \mathbf{v}_{t-1} + (1-\beta_2)\mathbf{g}_t^2$      $\triangleright$ entrywise multiplication
7:     $\widehat{\mathbf{m}}_t \leftarrow (1-\beta_1^t)^{-1}\mathbf{m}_t$
8:     $\widehat{\mathbf{v}}_t \leftarrow (1-\beta_2^t)^{-1}\mathbf{v}_t$
9:     $\mathbf{x}_{t+1} \leftarrow \mathbf{x}_t - \eta_t \left( \frac{\widehat{\mathbf{m}}_t}{\sqrt{\widehat{\mathbf{v}}_t}+\epsilon\mathbf{1}} + \lambda \mathbb{I}(\mathbf{m}_t \mathbf{x}_t \geq \mathbf{0})\mathbf{x}_t \right)$      $\triangleright$ entrywise operations
10:     $t \leftarrow t+1$
11: **until** stopping criterion is met
12: **return** optimized parameters $\mathbf{x}_t$

---

## C FIXED-POINT ANALYSIS

Revisiting the fixed-point analysis in Section 2.2 for ADAMW, suppose the trajectory of ADAMW converges to a fixed point $(\mathbf{x}^\star, \widehat{\mathbf{m}}^\star, \widehat{\mathbf{v}}^\star)$, so that $\widehat{\mathbf{m}}^\star = \nabla f(\mathbf{x}^\star)$ and $\widehat{\mathbf{v}}^\star = \nabla f(\mathbf{x}^\star)^2$. Passing to the

limit $\epsilon \searrow 0$, the fixed-point condition gives

$$\frac{\nabla f(\mathbf{x}^\star)}{|\nabla f(\mathbf{x})^\star| + \epsilon \mathbf{1}} + \lambda \mathbf{x}^\star \to \operatorname{sgn}(\nabla f(\mathbf{x}^\star)) + \lambda \mathbf{x}^\star = \mathbf{0}.$$

Taking inner products with $\nabla f(\mathbf{x}^\star)$ yields $\|\nabla f(\mathbf{x}^\star)\|_1 + \langle \lambda \mathbf{x}^\star, \nabla f(\mathbf{x}^\star) \rangle = 0$, which shows that $\mathbf{x}^\star$ is a Karush–Kuhn–Tucker (KKT) point of the constrained optimization problem

$$\min_{\mathbf{x} \in \mathbb{R}^d} f(\mathbf{x}) \quad \text{s.t.} \quad \|\mathbf{x}\|_\infty \leq \frac{1}{\lambda} \tag{3}$$

by Lemma 3.8 of Xie & Li (2024). Intuitively, ADAMW normalizes the gradient to its coordinate-wise sign at stationarity and then balances it against the linear pull of the decoupled weight decay, which enforces a box constraint on the parameters. Xie & Li (2024) formalize this intuition and show that whenever the iterates of ADAMW converge, the limit point is a KKT point of the box-constrained problem (3). However, this guarantee holds only under the assumption of convergence, and ADAMW is not known to converge in general.

We remark that we can adapt this argument for another, more heuristic insight into why optimizers with cautious weight decay perform unbiased optimization. Suppose ADAM with cautious weight decay reaches a fixed point, so that

$$\frac{\nabla f(\mathbf{x}^\star)}{|\nabla f(\mathbf{x}^\star)| + \epsilon \mathbf{1}} = -\lambda \mathbb{I}(\nabla f(\mathbf{x}^\star)\mathbf{x}^\star \geq \mathbf{0})\mathbf{x}^\star.$$

For a fixed point of LION-$\mathcal{K}$ with cautious weight decay, we have

$$-\nabla \mathcal{K}(-\nabla f(\mathbf{x}^\star)) = \lambda \mathbb{I}(\nabla \mathcal{K}(-\nabla f(\mathbf{x}^\star))\mathbf{x}^\star \leq \mathbf{0})\mathbf{x}^\star.$$

In either situation, casework on the signs of the update and $\mathbf{x}^\star$ shows that both sides must be $\mathbf{0}$. It follows that $\nabla f(\mathbf{x}^\star) = \mathbf{0}$ for ADAM and $\nabla \mathcal{K}(-\nabla f(\mathbf{x}^\star)) = \mathbf{0}$ for LION-$\mathcal{K}$, and if $\mathcal{K}$ is a convex function that achieves a unique minimum at $\mathbf{0}$ (e.g. a norm), then this condition becomes $\nabla f(\mathbf{x}^\star) = \mathbf{0}$ as well. Hence, the fixed-point analysis suggests that ADAM and LION-$\mathcal{K}$ with cautious weight decay find a stationary point of the original objective $f$.

## D    LYAPUNOV FUNCTIONS

Throughout this section, vector variables are implicitly dependent on $t$ when clear from context, and we drop the subscript for notational simplicity.

### D.1    SGD

SGD with cautious weight decay admits the continuous-time dynamics

$$\dot{\mathbf{x}} = -\nabla f(\mathbf{x}) - \lambda \mathbb{I}(\nabla f(\mathbf{x})\mathbf{x} \geq \mathbf{0})\mathbf{x},$$

which has a Lyapunov function $\mathcal{H}(\mathbf{x}) = f(\mathbf{x})$, since

$$\frac{d\mathcal{H}}{dt} = \langle \nabla f(\mathbf{x}), -\nabla f(\mathbf{x}) - \lambda \mathbb{I}(\nabla f(\mathbf{x})\mathbf{x} \geq \mathbf{0})\mathbf{x} \rangle = -\|\nabla f(\mathbf{x})\|_2^2 - \lambda \left\|(\nabla f(\mathbf{x})\mathbf{x})^+\right\|_1 \leq 0.$$

### D.2    SGD WITH MOMENTUM

When SGD is equipped with momentum (Sutskever et al., 2013) and cautious weight decay, the continuous-time dynamics becomes

$$\dot{\mathbf{x}} = -\mathbf{m} - \lambda \mathbb{I}(\mathbf{mx} \geq \mathbf{0})\mathbf{x}$$
$$\dot{\mathbf{m}} = \beta(\nabla f(\mathbf{x}) - \mathbf{m}),$$

which has a Lyapunov function

$$\mathcal{H}(\mathbf{x}, \mathbf{m}) = \beta f(\mathbf{x}) + \frac{1}{2}\|\mathbf{m}\|_2^2 + \lambda \left\|(\mathbf{mx})^+\right\|_1,$$

since

$$\frac{d\mathcal{H}}{dt} = \langle \beta \nabla f(\mathbf{x}) + \lambda \mathbb{I}(\mathbf{mx} \geq \mathbf{0})\mathbf{m}, -\mathbf{m} - \lambda \mathbb{I}(\mathbf{mx} \geq \mathbf{0})\mathbf{x} \rangle + \langle \mathbf{m} + \lambda \mathbb{I}(\mathbf{mx} \geq \mathbf{0})\mathbf{x}, \beta(\nabla f(\mathbf{x}) - \mathbf{m}) \rangle$$
$$= -\langle \lambda \mathbb{I}(\mathbf{mx} \geq \mathbf{0}) + \beta \mathbf{1}, \mathbf{m}^2 \rangle - \lambda(\beta + \lambda)\left\|(\mathbf{mx})^+\right\|_1 \leq 0.$$

### D.3 LION-$\mathcal{K}$

We assume that $\mathcal{K}$ is convex and satisfies $\mathrm{sgn}(\nabla\mathcal{K}(\mathbf{m})) = \mathrm{sgn}(\mathbf{m})$ for all $\mathbf{m} \in \mathbb{R}^d$. This assumption is mild and that holds for every example of $\mathcal{K}$ given by Chen et al. (2024).

The continuous-time dynamics of LION-$\mathcal{K}$ without gradient enhancement is given by

$$\begin{aligned} \dot{\mathbf{x}} &= \nabla\mathcal{K}(\mathbf{m}) - \lambda\mathbf{x} \\ \dot{\mathbf{m}} &= -\alpha\nabla f(\mathbf{x}) - \gamma\mathbf{m}. \end{aligned} \tag{4}$$

Chen et al. (2024) showed that this system has a Lyapunov function

$$\mathcal{H}(\mathbf{x}, \mathbf{m}) = \alpha f(\mathbf{x}) + \frac{\gamma}{\lambda}\mathcal{K}^*(\lambda\mathbf{x}) + \mathcal{K}^*(\lambda\mathbf{x}) + \mathcal{K}(\mathbf{m}) - \langle\mathbf{m}, \lambda\mathbf{x}\rangle,$$

thereby elucidating the origin of the $\mathcal{K}^*(\lambda\mathbf{x})$ regularization term. However, when equipped with cautious weight decay, (4) becomes

$$\begin{aligned} \dot{\mathbf{x}} &= \nabla\mathcal{K}(\mathbf{m}) - \lambda\mathbb{I}(\mathbf{mx} \leq \mathbf{0})\mathbf{x} \\ \dot{\mathbf{m}} &= -\alpha\nabla f(\mathbf{x}) - \gamma\mathbf{m} \end{aligned} \tag{5}$$

and admits a Lyapunov function

$$\mathcal{H}(\mathbf{x}, \mathbf{m}) = \alpha f(\mathbf{x}) + \mathcal{K}(\mathbf{m}) + \lambda\left\|(-\mathbf{mx})^+\right\|_1, \tag{6}$$

which corresponds to optimizing the original objective $f$. To see that (6) is a Lyapunov function for (5), note that

$$\begin{aligned} \frac{\mathrm{d}\mathcal{H}}{\mathrm{d}t} &= \langle\alpha\nabla f(\mathbf{x}) - \lambda\mathbb{I}(\mathbf{mx} \leq \mathbf{0})\mathbf{m}, \nabla\mathcal{K}(\mathbf{m}) - \lambda\mathbb{I}(\mathbf{mx} \leq \mathbf{0})\mathbf{x}\rangle \\ &\quad + \langle\nabla\mathcal{K}(\mathbf{m}) - \lambda\mathbb{I}(\mathbf{mx} \leq \mathbf{0})\mathbf{x}, -\alpha\nabla f(\mathbf{x}) - \gamma\mathbf{m}\rangle \\ &= -\langle\nabla\mathcal{K}(\mathbf{m}) - \lambda\mathbb{I}(\mathbf{mx} \leq \mathbf{0})\mathbf{x}, (\lambda\mathbb{I}(\mathbf{mx} \leq \mathbf{0}) + \gamma\mathbf{1})\mathbf{m}\rangle \\ &= -\langle\lambda\mathbb{I}(\mathbf{mx} \leq \mathbf{0}) + \gamma\mathbf{1}, \nabla\mathcal{K}(\mathbf{m})\mathbf{m}\rangle - \lambda(\lambda + \gamma)\left\|(-\mathbf{mx})^+\right\|_1 \leq 0. \end{aligned}$$

### D.4 ADAM

The continuous-time limit of ADAM with cautious weight decay yields the system of ordinary differential equations (cf. Barakat & Bianchi (2021))

$$\begin{aligned} \dot{\mathbf{x}} &= -\frac{(1 - \exp(-\alpha t))^{-1}\mathbf{m}}{\sqrt{(1 - \exp(-\gamma t))^{-1}\mathbf{v} + \epsilon\mathbf{1}}} - \lambda\mathbb{I}(\mathbf{mx} \geq \mathbf{0})\mathbf{x} \\ \dot{\mathbf{m}} &= \alpha(\nabla f(\mathbf{x}) - \mathbf{m}) \\ \dot{\mathbf{v}} &= \gamma(\nabla f(\mathbf{x})^2 - \mathbf{v}). \end{aligned} \tag{7}$$

We assume that $0 < \gamma \leq 4\alpha$, which is satisfied by standard implementations of ADAM in practice. This system admits the Lyapunov function

$$\mathcal{H}(\mathbf{x}, \mathbf{m}, \mathbf{v}, t) = \alpha f(\mathbf{x}) + \left\|\frac{\alpha_t\mathbf{m}^2}{2(\sqrt{\gamma_t\mathbf{v}} + \epsilon\mathbf{1})}\right\|_1 + \lambda\left\|(\mathbf{mx})^+\right\|_1, \tag{8}$$

where

$$\alpha_t := (1 - \exp(-\alpha t))^{-1} \quad \text{and} \quad \gamma_t := (1 - \exp(-\gamma t))^{-1}.$$

To see that (8) is a Lyapunov function for (7), note that $\mathcal{H}$ is lower bounded by $\alpha f^\star$ and

$$\frac{\mathrm{d}\mathcal{H}}{\mathrm{d}t} = \langle \nabla_{\mathbf{x}}\mathcal{H}, \dot{\mathbf{x}} \rangle + \langle \nabla_{\mathbf{m}}\mathcal{H}, \dot{\mathbf{m}} \rangle + \langle \nabla_{\mathbf{v}}\mathcal{H}, \dot{\mathbf{v}} \rangle + \frac{\partial \mathcal{H}}{\partial t}$$

$$= \left\langle \alpha \nabla f(\mathbf{x}) + \lambda \mathbb{I}(\mathbf{mx} \geq \mathbf{0})\mathbf{m}, -\frac{\alpha_t \mathbf{m}}{\sqrt{\gamma_t \mathbf{v}} + \epsilon \mathbf{1}} - \lambda \mathbb{I}(\mathbf{mx} \geq \mathbf{0})\mathbf{x} \right\rangle$$

$$+ \left\langle \frac{\alpha_t \mathbf{m}}{\sqrt{\gamma_t \mathbf{v}} + \epsilon \mathbf{1}} + \lambda \mathbb{I}(\mathbf{mx} \geq \mathbf{0})\mathbf{x}, \alpha(\nabla f(\mathbf{x}) - \mathbf{m}) \right\rangle - \left\langle \frac{\alpha_t \sqrt{\gamma_t}\mathbf{m}^2}{4\sqrt{\mathbf{v}}\left(\sqrt{\gamma_t \mathbf{v}} + \epsilon \mathbf{1}\right)^2}, \gamma(\nabla f(\mathbf{x})^2 - \mathbf{v}) \right\rangle$$

$$- \left\langle \frac{\mathbf{m}^2}{2} \cdot \frac{2\alpha \exp(-\alpha t)(\sqrt{\gamma_t \mathbf{v}} + \epsilon \mathbf{1}) - \alpha_t^{-1}\gamma \exp(-\gamma t)\gamma_t \sqrt{\gamma_t \mathbf{v}}}{2\left(\alpha_t^{-1}(\sqrt{\gamma_t \mathbf{v}} + \epsilon \mathbf{1})\right)^2}, \mathbf{1} \right\rangle$$

$$= -\left\langle (\alpha \mathbf{1} + \lambda \mathbb{I}(\mathbf{mx} \geq \mathbf{0}))\frac{\alpha_t \mathbf{m}^2}{\sqrt{\gamma_t \mathbf{v}} + \epsilon \mathbf{1}} + \lambda(\alpha + \lambda)(\mathbf{mx})^+ + \frac{\alpha_t \gamma \sqrt{\gamma_t}\mathbf{m}^2 \nabla f(\mathbf{x})^2}{4\sqrt{\mathbf{v}}\left(\sqrt{\gamma_t \mathbf{v}} + \epsilon \mathbf{1}\right)^2}, \mathbf{1} \right\rangle$$

$$+ \left\langle \frac{\alpha_t \gamma \mathbf{m}^2 \sqrt{\gamma_t \mathbf{v}}}{4\left(\sqrt{\gamma_t \mathbf{v}} + \epsilon \mathbf{1}\right)^2}, \mathbf{1} \right\rangle - \left\langle \frac{\mathbf{m}^2}{2} \cdot \frac{2\alpha \exp(-\alpha t)(\sqrt{\gamma_t \mathbf{v}} + \epsilon \mathbf{1}) - \alpha_t^{-1}\gamma \exp(-\gamma t)\gamma_t \sqrt{\gamma_t \mathbf{v}}}{2\left(\alpha_t^{-1}(\sqrt{\gamma_t \mathbf{v}} + \epsilon \mathbf{1})\right)^2}, \mathbf{1} \right\rangle$$

$$\leq \left\langle \left(\frac{\gamma}{4} - \alpha\right)\mathbf{1} - \lambda \mathbb{I}(\mathbf{mx} \geq \mathbf{0}), \frac{\alpha_t \mathbf{m}^2}{\sqrt{\gamma_t \mathbf{v}} + \epsilon \mathbf{1}} \right\rangle - \left\langle \frac{\alpha_t(2\alpha_t \alpha \exp(-\alpha t) - \gamma_t \gamma \exp(-\gamma t))\mathbf{m}^2}{4(\sqrt{\gamma_t \mathbf{v}} + \epsilon \mathbf{1})}, \mathbf{1} \right\rangle$$

$$= \left\langle \left(\frac{\gamma}{4} - \alpha - \frac{\alpha}{2(\exp(\alpha t) - 1)} + \frac{\gamma}{4(\exp(\gamma t) - 1)}\right)\mathbf{1} - \lambda \mathbb{I}(\mathbf{mx} \geq \mathbf{0}), \frac{\alpha_t \mathbf{m}^2}{\sqrt{\gamma_t \mathbf{v}} + \epsilon \mathbf{1}} \right\rangle$$

$$\leq 0,$$

where the first inequality drops some nonpositive terms and uses $\sqrt{\gamma_t \mathbf{v}} \leq \sqrt{\gamma_t \mathbf{v}} + \epsilon \mathbf{1}$ and the second inequality uses

$$\frac{\gamma}{4} - \alpha - \frac{\alpha}{2(\exp(\alpha t) - 1)} + \frac{\gamma}{4(\exp(\gamma t) - 1)} \leq 0$$

for $0 < \gamma \leq 4\alpha$ and $t > 0$.

**Remark 1.** *Cautious weight decay can be seen as an attempt to fix the asymptotic instability of* ADAMW *via a Lyapunov function. Consider the simplified continuous-time* ADAMW *dynamics*

$$\dot{\mathbf{x}} = -\frac{\mathbf{m}}{\sqrt{\mathbf{v}}} - \lambda \mathbf{x}$$

$$\dot{\mathbf{m}} = \nabla f(\mathbf{x}) - \mathbf{m} \tag{9}$$

$$\dot{\mathbf{v}} = \nabla f(\mathbf{x})^2 - \mathbf{v}$$

*and the function*

$$\mathcal{H}(\mathbf{x}, \mathbf{m}, \mathbf{v}) = f(\mathbf{x}) + \left\| \frac{\mathbf{m}^2}{2\sqrt{\mathbf{v}}} \right\|_1 + \langle \mathbf{m}, \lambda \mathbf{x} \rangle.$$

*By straightforward computation,*

$$\frac{\mathrm{d}\mathcal{H}}{\mathrm{d}t} = \left\langle \nabla f(\mathbf{x}) + \lambda \mathbf{m}, -\frac{\mathbf{m}}{\sqrt{\mathbf{v}}} - \lambda \mathbf{x} \right\rangle + \left\langle \frac{\mathbf{m}}{\sqrt{\mathbf{v}}} + \lambda \mathbf{x}, \nabla f(\mathbf{x}) - \mathbf{m} \right\rangle + \left\langle -\frac{\mathbf{m}^2}{4\mathbf{v}^{\frac{3}{2}}}, \nabla f(\mathbf{x})^2 - \mathbf{v} \right\rangle$$

$$= -\left\langle \left(\lambda + \frac{3}{4}\right)\frac{\mathbf{m}^2}{\sqrt{\mathbf{v}}} + \lambda(\lambda + 1)\mathbf{mx} + \frac{\mathbf{m}^2 \nabla f(\mathbf{x})^2}{4\mathbf{v}^{\frac{3}{2}}}, \mathbf{1} \right\rangle$$

$$= -\left(\lambda + \frac{3}{4}\right)\left\| \frac{\mathbf{m}^2}{\sqrt{\mathbf{v}}} \right\|_1 - \lambda(\lambda + 1)\langle \mathbf{m}, \mathbf{x} \rangle - \frac{1}{4}\left\| \frac{\mathbf{m}^2 \nabla f(\mathbf{x})^2}{\mathbf{v}^{\frac{3}{2}}} \right\|_1.$$

*Note that $\mathcal{H}$ is not guaranteed to be lower bounded and $-\frac{\mathrm{d}\mathcal{H}}{\mathrm{d}t}$ is not guaranteed to be nonnegative, since $\langle \mathbf{m}, \mathbf{x} \rangle$ has unknown sign. This motivates the introduction of a mask $\mathbb{I}(\mathbf{mx} \geq \mathbf{0})$ to the weight decay term and a slight adjustment to $\mathcal{H}$ so that the result is a Lyapunov function for (9).*

**Remark 2.** *For expositional clarity, we treat the ODEs and Lyapunov candidates in this section as smooth, even though the dynamics include the discontinuous indicator function $\mathbb{I}(\mathbf{ux} \geq \mathbf{0})$. A fully rigorous analysis can be developed by interpreting the systems in the sense of differential inclusions, specifically, using Filippov's framework (Filippov, 1988), and by applying specialized tools from nonsmooth Lyapunov stability theory to obtain convergence guarantees (Shevitz & Paden, 1994; Bacciotti & Ceragioli, 1999).*

# E    CONVERGENCE RATE OF ADAM WITH CAUTIOUS WEIGHT DECAY

In this section, we show that under the following assumptions, ADAM with cautious weight decay (Algorithm 6) achieves a convergence rate on the squared gradient norm and an additional stationarity measure.

**Assumption 1** (Smoothness). *$f$ is $L$-smooth and lower bounded by a finite constant $f^\star$.*

**Assumption 2** (Bounded variance). *The stochastic gradient $\mathbf{g}_t$ satisfies*

$$\mathbb{E}[\mathbf{g}_t \mid \mathbf{x}_t] = \nabla f(\mathbf{x}_t) \quad and \quad \mathrm{Var}(\mathbf{g}_t) = \mathbb{E}\left[\|\mathbf{g}_t - \nabla f(\mathbf{x}_t)\|_2^2 \mid \mathbf{x}_t\right] \leq \frac{\sigma^2}{n_{\mathrm{batch}}},$$

*where $\sigma$ is a constant and $n_{\mathrm{batch}}$ denotes the batch size.*

**Assumption 3** (Bounded iterates and bounded gradients). *There exist constants $R$ and $G$ such that $\|\mathbf{x}_t\|_\infty \leq R$ and $\|\mathbf{g}_t\|_\infty \leq G$ a.s. for all $t \in \mathbb{N}$.*

Assumptions 1 and 2 are standard and often used in the analysis of stochastic gradient algorithms (Ghadimi & Lan, 2013; Barakat & Bianchi, 2021; Défossez et al., 2022; Arjevani et al., 2023). Assumption 3 can be justified using the Lyapunov function (8) if $f$ is additionally assumed to be coercive, since a Robbins–Siegmund argument with sufficiently small $\eta$ shows that the optimizer states remain in a compact sublevel set of $\mathcal{H}$ a.s. For the sake of clarity, here we take it as an explicit assumption. Similar assumptions have often been used for the analysis of ADAM-style algorithms (Kingma & Ba, 2015; Reddi et al., 2018; Zaheer et al., 2018; Chen et al., 2019; Défossez et al., 2022; Chen et al., 2022).

**Theorem 1.** *Under Assumptions 1, 2, and 3, let $0 \leq \beta_1 \leq \beta_2 < 1$, $\lambda \geq 0$, $\epsilon > 0$, and $\eta_t = \eta > 0$, and suppose $\mathbf{x}_t$ is updated using Algorithm 6. Then for all $T \in \mathbb{N}$,*

$$\frac{1}{T} \sum_{t \in [T]} \mathbb{E}\left[\|\nabla f(\mathbf{x}_t)\|_2^2 + \lambda \left\|(\nabla f(\mathbf{x}_t)\mathbf{x}_t)^+\right\|_1\right] \leq \frac{K_1}{\eta T} + \frac{K_2}{T} + K_3\eta + \frac{K_4\sigma}{\sqrt{n_{\mathrm{batch}}}},$$

*where $K_1$, $K_2$, $K_3$, and $K_4$ are constants depending only on $L$, $R$, $G$, $d$, $\epsilon$, $\lambda$, $\beta_1$, $\beta_2$, and $f(\mathbf{x}_1) - f^\star$.*

**Remark 3.** *The first term on the left-hand side, $\|\nabla f(\mathbf{x}_t)\|_2^2$, reflects how much $f$ is optimized, while the second term, $\|(\nabla f(\mathbf{x}_t)\mathbf{x}_t)^+\|_1$, reflects the degree of conflict between the objective $f$ and the parameter magnitudes. If $\nabla f(\mathbf{x}_t)\mathbf{x}_t \gg \mathbf{0}$, then there is room to jointly decrease both $f$ and the magnitudes. Thus, a small value of $\|(\nabla f(\mathbf{x}_t)\mathbf{x}_t)^+\|_1$ indicates that the optimizer has reached a state where it is difficult to further decrease $f$ and shrink the magnitudes simultaneously. This corresponds to convergence toward a Pareto front, where trade-offs between the two objectives become unavoidable.*

**Remark 4.** *In the setting of Theorem 1, let $T \in \mathbb{N}$ and $\eta = \Theta\left(\frac{1}{\sqrt{T}}\right)$. Then*

$$\frac{1}{T} \sum_{t \in [T]} \mathbb{E}\left[\|\nabla f(\mathbf{x}_t)\|_2^2 + \lambda \left\|(\nabla f(\mathbf{x}_t)\mathbf{x}_t)^+\right\|_1\right] = O\left(\frac{1}{\sqrt{T}} + \frac{\sigma}{\sqrt{n_{\mathrm{batch}}}}\right).$$

*An $O(T^{-\frac{1}{2}})$ bound can be obtained by making the unrealistic assumption $n_{\mathrm{batch}} = \Theta(T)$. However, even without this assumption, the stated bound is of theoretical interest. For additional discussion, see Bernstein et al. (2018); Zaheer et al. (2018); Chen et al. (2024).*

**Lemma 1.** *For all $t \in \mathbb{N}$,*

$$\left\|\frac{\widehat{\mathbf{m}}_t}{\sqrt{\widehat{\mathbf{v}}_t} + \epsilon\mathbf{1}}\right\|_\infty \leq \sqrt{\frac{1 - \beta_1}{1 - \beta_2}} =: C.$$

*Proof.* It suffices to work in an arbitrary coordinate $i$. Let $m := [\widehat{\mathbf{m}}_t]_i$, $v := [\widehat{\mathbf{v}}_t]_i$, and $g_t := [\mathbf{g}_t]_i$. By expanding the update rules for $m$ and $v$, we obtain

$$m = \frac{1 - \beta_1}{1 - \beta_1^t} \sum_{k \in [t]} \beta_1^{t-k} g_k \quad and \quad v = \frac{1 - \beta_2}{1 - \beta_2^t} \sum_{k \in [t]} \beta_2^{t-k} g_k^2.$$

Now by Cauchy–Schwarz,

$$\frac{m^2}{v} \leq \frac{(1-\beta_1)^2}{(1-\beta_1^t)^2} \cdot \frac{1-\beta_2^t}{1-\beta_2} \cdot \sum_{k\in[t]}\left(\frac{\beta_1^2}{\beta_2}\right)^{t-k} \leq \frac{(1-\beta_1)^2}{(1-\beta_1^t)^2} \cdot \frac{1-\beta_2^t}{1-\beta_2} \cdot \sum_{k\in[t]}\beta_1^{t-k}$$

$$= \frac{(1-\beta_1)^2}{(1-\beta_1^t)^2} \cdot \frac{1-\beta_2^t}{1-\beta_2} \cdot \frac{1-\beta_1^t}{1-\beta_1} = \frac{1-\beta_1}{1-\beta_2} \cdot \frac{1-\beta_2^t}{1-\beta_1^t} \leq \frac{1-\beta_1}{1-\beta_2}.$$

The conclusion follows from

$$\frac{m}{\sqrt{v}+\epsilon} \leq \frac{m}{\sqrt{v}} \leq \sqrt{\frac{1-\beta_1}{1-\beta_2}}.$$

$\square$

**Fact 1** (Lemma F.1, Bernstein et al. (2018)). *For all $t \in \mathbb{N}$, $i \in [d]$, and $\alpha_1, \alpha_2, \ldots, \alpha_t \in \mathbb{R}$,*

$$\mathbb{E}\left[\left(\sum_{k\in[t]}\alpha_k([\mathbf{g}_k]_i - [\nabla f(\mathbf{x}_k)]_i)\right)^2\right] \leq \frac{\sigma^2}{n_{\text{batch}}}\sum_{k\in[t]}\alpha_k^2.$$

**Lemma 2.** *For all $t \in \mathbb{N}$,*

$$\mathbb{E}[\|\nabla f(\mathbf{x}_t) - \mathbf{m}_t\|_1] \leq \beta_1^t Gd + \frac{\beta_1\eta Ld(C+\lambda R)}{1-\beta_1} + \frac{\sigma d}{\sqrt{n_{\text{batch}}(1+\beta_1)}}.$$

*Proof.* Note that

$$\mathbf{m}_t - \nabla f(\mathbf{x}_t) = -\beta_1^t\nabla f(\mathbf{x}_1) + \sum_{k\in[t-1]}\beta_1^{t-k}(\nabla f(\mathbf{x}_k) - \nabla f(\mathbf{x}_{k+1})) + (1-\beta_1)\sum_{k\in[t]}\beta_1^{t-k}(\mathbf{g}_k - \nabla f(\mathbf{x}_k)).$$

$$(10)$$

By smoothness, Lemma 1, and Assumption 3, we have

$$\|\nabla f(\mathbf{x}_k) - \nabla f(\mathbf{x}_{k+1})\|_1 \leq \sqrt{d}\|\nabla f(\mathbf{x}_k) - \nabla f(\mathbf{x}_{k+1})\|_2 \leq L\sqrt{d}\|\mathbf{x}_{k+1} - \mathbf{x}_k\|_2 \leq \eta Ld(C+\lambda R).$$

$$(11)$$

By Jensen's inequality and Fact 1,

$$\mathbb{E}\left[\left|\sum_{k\in[t]}\beta_1^{t-k}([\mathbf{g}_k]_i - [\nabla f(\mathbf{x}_k)]_i)\right|\right] \leq \sqrt{\mathbb{E}\left[\left(\sum_{k\in[t]}\beta_1^{t-k}([\mathbf{g}_k]_i - [\nabla f(\mathbf{x}_k)]_i)\right)^2\right]}$$

$$\leq \sqrt{\frac{\sigma^2}{n_{\text{batch}}}\sum_{k\in[t]}(\beta_1^2)^{t-k}} \leq \frac{\sigma}{\sqrt{n_{\text{batch}}(1-\beta_1^2)}}.$$

$$(12)$$

Taking $\mathbb{E}[\|\cdot\|_1]$ of (10) and applying (11) and (12),

$$\mathbb{E}[\|\nabla f(\mathbf{x}_t) - \mathbf{m}_t\|_1] \leq \beta_1^t\|\nabla f(\mathbf{x}_1)\|_1 + \frac{\beta_1\eta Ld(C+\lambda R)}{1-\beta_1} + (1-\beta_1)\mathbb{E}\left[\left\|\sum_{k\in[t]}\beta_1^{t-k}(\mathbf{g}_k - \nabla f(\mathbf{x}_k))\right\|_1\right]$$

$$\leq \beta_1^t Gd + \frac{\beta_1\eta Ld(C+\lambda R)}{1-\beta_1} + \frac{\sigma d}{\sqrt{n_{\text{batch}}(1+\beta_1)}},$$

as desired. $\square$

**Lemma 3.** *For all $t \in \mathbb{N}$,*

$$\mathbb{E}\left[-\left\langle\nabla f(\mathbf{x}_t), \frac{\mathbf{m}_t}{\sqrt{\widehat{\mathbf{v}}_t}+\epsilon\mathbf{1}}\right\rangle\right] \leq -\frac{\mathbb{E}\left[\|\nabla f(\mathbf{x}_t)\|_2^2\right]}{G+\epsilon} + \frac{\beta_1^t G^2 d}{\epsilon} + \frac{\beta_1\eta GLd(C+\lambda R)}{(1-\beta_1)\epsilon} + \frac{\sigma Gd}{\epsilon\sqrt{n_{\text{batch}}(1+\beta_1)}}.$$

*Proof.* We have

$$-\left\langle \nabla f(\mathbf{x}_t), \frac{\mathbf{m}_t}{\sqrt{\widehat{\mathbf{v}}_t} + \epsilon \mathbf{1}} \right\rangle = \left\langle \frac{\nabla f(\mathbf{x}_t)}{\sqrt{\widehat{\mathbf{v}}_t} + \epsilon \mathbf{1}}, \nabla f(\mathbf{x}_t) - \mathbf{m}_t - \nabla f(\mathbf{x}_t) \right\rangle$$

$$\leq -\frac{1}{G + \epsilon} \|\nabla f(\mathbf{x}_t)\|_2^2 + \left\langle \frac{\nabla f(\mathbf{x}_t)}{\sqrt{\widehat{\mathbf{v}}_t} + \epsilon \mathbf{1}}, \nabla f(\mathbf{x}_t) - \mathbf{m}_t \right\rangle$$

$$\leq -\frac{1}{G + \epsilon} \|\nabla f(\mathbf{x}_t)\|_2^2 + \left\| \frac{\nabla f(\mathbf{x}_t)}{\sqrt{\widehat{\mathbf{v}}_t} + \epsilon \mathbf{1}} \right\|_\infty \|\nabla f(\mathbf{x}_t) - \mathbf{m}_t\|_1$$

The result follows by $\left\| \frac{\nabla f(\mathbf{x}_t)}{\sqrt{\widehat{\mathbf{v}}_t} + \epsilon \mathbf{1}} \right\|_\infty \leq \frac{G}{\epsilon}$ and Lemma 2 . $\qquad\square$

**Lemma 4.** *For all $m, g, x \in \mathbb{R}$,*

$$|(\mathbb{I}(mx \geq 0) - \mathbb{I}(gx \geq 0))x| \leq \mathbb{I}(mg \leq 0)|x|.$$

*Proof.* If $x = 0$, then the inequality is trivially valid, so suppose $x \neq 0$. We proceed by casework on the sign of $mg$.

If $mg > 0$, then $m$ and $g$ have the same sign, and the conditions $mx \geq 0$ and $gx \geq 0$ are equivalent. Thus $\mathbb{I}(mx \geq 0) - \mathbb{I}(gx \geq 0) = 0$, and the inequality holds.

If $mg \leq 0$, then $\mathbb{I}(mg \leq 0) = 1$. It remains to show $|(\mathbb{I}(mx \geq 0) - \mathbb{I}(gx \geq 0))x| \leq |x|$, which follows upon realizing $\mathbb{I}(mx \geq 0) - \mathbb{I}(gx \geq 0) \in \{-1, 0, 1\}$. $\qquad\square$

**Lemma 5.** *For all $t \in \mathbb{N}$,*

$$\mathbb{E}[-\langle \nabla f(\mathbf{x}_t), \mathbb{I}(\mathbf{m}_t \mathbf{x}_t \geq \mathbf{0})\mathbf{x}_t \rangle] \leq -\mathbb{E}\left[\left\| (\nabla f(\mathbf{x}_t)\mathbf{x}_t)^+ \right\|_1\right] + \beta_1^t GRd + \frac{\beta_1 \eta LRd(C + \lambda R)}{1 - \beta_1} + \frac{\sigma Rd}{\sqrt{n_{\text{batch}}(1 + \beta_1)}}.$$

*Proof.* We have

$$-\langle \nabla f(\mathbf{x}_t), \mathbb{I}(\mathbf{m}_t \mathbf{x}_t \geq \mathbf{0})\mathbf{x}_t \rangle = -\langle \nabla f(\mathbf{x}_t), \mathbb{I}(\mathbf{x}_t \nabla f(\mathbf{x}_t) \geq \mathbf{0})\mathbf{x}_t + (\mathbb{I}(\mathbf{m}_t \mathbf{x}_t \geq \mathbf{0}) - \mathbb{I}(\mathbf{x}_t \nabla f(\mathbf{x}_t) \geq \mathbf{0}))\mathbf{x}_t \rangle$$

$$= \langle \nabla f(\mathbf{x}_t), (\mathbb{I}(\mathbf{x}_t \nabla f(\mathbf{x}_t) \geq \mathbf{0}) - \mathbb{I}(\mathbf{m}_t \mathbf{x}_t \geq \mathbf{0}))\mathbf{x}_t \rangle - \left\| (\nabla f(\mathbf{x}_t)\mathbf{x}_t)^+ \right\|_1$$

$$\leq \langle |\nabla f(\mathbf{x}_t)|, |(\mathbb{I}(\mathbf{x}_t \nabla f(\mathbf{x}_t) \geq \mathbf{0}) - \mathbb{I}(\mathbf{m}_t \mathbf{x}_t \geq \mathbf{0}))\mathbf{x}_t| \rangle - \left\| (\nabla f(\mathbf{x}_t)\mathbf{x}_t)^+ \right\|_1$$

$$\leq \langle |\nabla f(\mathbf{x}_t)|, \mathbb{I}(\mathbf{m}_t \nabla f(\mathbf{x}_t) \leq \mathbf{0})|\mathbf{x}_t| \rangle - \left\| (\nabla f(\mathbf{x}_t)\mathbf{x}_t)^+ \right\|_1,$$

$$\tag{13}$$

where the fourth line uses Lemma 4. Taking the expectation of (13) conditioned on $\mathbf{x}_t$ and expanding the inner product,

$$\mathbb{E}[\langle |\nabla f(\mathbf{x}_t)|, \mathbb{I}(\mathbf{m}_t \nabla f(\mathbf{x}_t) \leq \mathbf{0})|\mathbf{x}_t| \rangle \mid \mathbf{x}_t] = \langle |\nabla f(\mathbf{x}_t)|, \mathbb{E}[\mathbb{I}(\mathbf{m}_t \nabla f(\mathbf{x}_t) \leq \mathbf{0}) \mid \mathbf{x}_t]|\mathbf{x}_t| \rangle$$

$$= \sum_{i \in [d]} |[\nabla f(\mathbf{x}_t)]_i[\mathbf{x}_t]_i| \cdot \mathbb{E}[\mathbb{I}([\mathbf{m}_t]_i[\nabla f(\mathbf{x}_t)]_i \leq 0) \mid \mathbf{x}_t]$$

$$= \sum_{i \in [d]} |[\nabla f(\mathbf{x}_t)]_i[\mathbf{x}_t]_i| \cdot \Pr([\mathbf{m}_t]_i[\nabla f(\mathbf{x}_t)]_i \leq 0 \mid \mathbf{x}_t)$$

$$\leq \sum_{i \in [d]} |[\nabla f(\mathbf{x}_t)]_i[\mathbf{x}_t]_i| \cdot \Pr(|[\nabla f(\mathbf{x}_t)]_i - [\mathbf{m}_t]_i| \geq |[\nabla f(\mathbf{x}_t)]_i| \mid \mathbf{x}_t)$$

$$\leq \sum_{i \in [d]} |[\mathbf{x}_t]_i| \cdot \mathbb{E}[|[\nabla f(\mathbf{x}_t)]_i - [\mathbf{m}_t]_i| \mid \mathbf{x}_t]$$

$$\leq R \cdot \mathbb{E}[\|\nabla f(\mathbf{x}_t) - \mathbf{m}_t\|_1 \mid \mathbf{x}_t],$$

$$\tag{14}$$

where the fifth line uses Markov's inequality. Taking the expectation of (14) and applying Lemma 2,

$$\mathbb{E}[-\langle \nabla f(\mathbf{x}_t), \mathbb{I}(\mathbf{m}_t \mathbf{x}_t \geq \mathbf{0})\mathbf{x}_t \rangle] \leq -\mathbb{E}\left[\left\| (\nabla f(\mathbf{x}_t)\mathbf{x}_t)^+ \right\|_1\right] + \beta_1^t GRd + \frac{\beta_1 \eta LRd(C + \lambda R)}{1 - \beta_1} + \frac{\sigma Rd}{\sqrt{n_{\text{batch}}(1 + \beta_1)}},$$

as desired. $\qquad\square$

We are now ready to prove Theorem 1.

*Proof of Theorem 1.* Let

$$\Delta_t := f(\mathbf{x}_{t+1}) - f(\mathbf{x}_t) \quad \text{and} \quad \boldsymbol{\delta}_t := \frac{\widehat{\mathbf{m}}_t}{\sqrt{\widehat{\mathbf{v}}_t} + \epsilon \mathbf{1}} + \lambda \mathbb{I}(\mathbf{m}_t \mathbf{x}_t \geq \mathbf{0}) \mathbf{x}_t.$$

By smoothness,

$$
\begin{aligned}
\Delta_t &\leq \langle \nabla f(\mathbf{x}_t), \mathbf{x}_{t+1} - \mathbf{x}_t \rangle + \frac{L}{2} \|\mathbf{x}_{t+1} - \mathbf{x}_t\|_2^2 \\
&= -\eta \langle \nabla f(\mathbf{x}_t), \boldsymbol{\delta}_t \rangle + \frac{\eta^2 L}{2} \|\boldsymbol{\delta}_t\|_2^2 \\
&= -\eta \left\langle \nabla f(\mathbf{x}_t), \frac{\widehat{\mathbf{m}}_t}{\sqrt{\widehat{\mathbf{v}}_t} + \epsilon \mathbf{1}} \right\rangle - \eta \lambda \langle \nabla f(\mathbf{x}_t), \mathbb{I}(\mathbf{m}_t \mathbf{x}_t \geq \mathbf{0}) \mathbf{x}_t \rangle + \frac{\eta^2 L}{2} \|\boldsymbol{\delta}_t\|_2^2 \\
&= -\frac{\eta}{1 - \beta_1^t} \left\langle \nabla f(\mathbf{x}_t), \frac{\mathbf{m}_t}{\sqrt{\widehat{\mathbf{v}}_t} + \epsilon \mathbf{1}} \right\rangle - \eta \lambda \langle \nabla f(\mathbf{x}_t), \mathbb{I}(\mathbf{m}_t \mathbf{x}_t \geq \mathbf{0}) \mathbf{x}_t \rangle + \frac{\eta^2 L}{2} \|\boldsymbol{\delta}_t\|_2^2 .
\end{aligned}
\tag{15}
$$

Taking the expectation of (15) and applying Lemmas 1, 3, and 5,

$$
\begin{aligned}
\mathbb{E}[\Delta_t] &\leq \frac{\eta}{1 - \beta_1^t} \left( -\frac{\mathbb{E}\left[ \|\nabla f(\mathbf{x}_t)\|_2^2 \right]}{G + \epsilon} + \frac{\beta_1^t G^2 d}{\epsilon} + \frac{\beta_1 \eta G L d (C + \lambda R)}{(1 - \beta_1)\epsilon} + \frac{\sigma G d}{\epsilon \sqrt{n_{\text{batch}}(1 + \beta_1)}} \right) \\
&\quad + \eta \lambda \left( -\mathbb{E}\left[ \|(\nabla f(\mathbf{x}_t) \mathbf{x}_t)^+\|_1 \right] + \beta_1^t G R d + \frac{\beta_1 \eta L R d (C + \lambda R)}{1 - \beta_1} + \frac{\sigma R d}{\sqrt{n_{\text{batch}}(1 + \beta_1)}} \right) \\
&\quad + \eta^2 L (C^2 d + \lambda^2 R^2 d).
\end{aligned}
\tag{16}
$$

We can assume $G \geq \frac{1}{1 - \beta_1}$ without loss of generality. Rearranging (16), using $1 - \beta_1^t \leq 1$ and $(1 - \beta_1^t)(G + \epsilon) \geq 1$, summing over $T$ iterations, and dividing both sides by $T$ gives

$$
\begin{aligned}
\frac{1}{T} \sum_{t \in [T]} \mathbb{E}[\mathcal{S}(\mathbf{x}_t)] &\leq \frac{G + \epsilon}{\eta T}(f(\mathbf{x}_1) - f^\star) + \frac{G + \epsilon}{T} \sum_{t \in [T]} \frac{\beta_1^t G^2 d}{\epsilon} + \frac{\beta_1 \eta G L d (C + \lambda R)(G + \epsilon)}{(1 - \beta_1)\epsilon} \\
&\quad + \frac{\sigma G d (G + \epsilon)}{\epsilon \sqrt{n_{\text{batch}}(1 + \beta_1)}} + \frac{\lambda(G + \epsilon)}{T} \sum_{t \in [T]} \beta_1^t G R d + \frac{\lambda \sigma R d (G + \epsilon)}{\sqrt{n_{\text{batch}}(1 + \beta_1)}} \\
&\quad + \frac{\beta_1 \eta \lambda L R d (C + \lambda R)(G + \epsilon)}{1 - \beta_1} + \eta L (G + \epsilon)(C^2 d + \lambda^2 R^2 d) \\
&\leq \frac{K_1}{\eta T} + \frac{K_2}{T} + K_3 \eta + \frac{K_4 \sigma}{\sqrt{n_{\text{batch}}}},
\end{aligned}
$$

where the fourth line uses $\sum_{t \in [T]} \beta_1^t \leq \frac{\beta_1}{1 - \beta_1}$ and

$$
\begin{aligned}
\mathcal{S}(\mathbf{x}_t) &:= \|\nabla f(\mathbf{x}_t)\|_2^2 + \lambda \left\| (\nabla f(\mathbf{x}_t) \mathbf{x}_t)^+ \right\|_1 \\
K_1 &:= (G + \epsilon)(f(\mathbf{x}_1) - f^\star) \\
K_2 &:= \frac{\beta_1 G d (G + \epsilon)}{1 - \beta_1} \left( \frac{G}{\epsilon} + \lambda R \right) \\
K_3 &:= \frac{\beta_1 L d (C + \lambda R)(G + \epsilon)}{1 - \beta_1} \left( \frac{G}{\epsilon} + \lambda R \right) + L d (C^2 + \lambda^2 R^2)(G + \epsilon) \\
K_4 &:= \frac{d(G + \epsilon)}{\sqrt{1 + \beta_1}} \left( \frac{G}{\epsilon} + \lambda R \right).
\end{aligned}
$$

$\square$

Table 8: Hyperparameter configurations for the different model sizes. All models use an expansion factor of 8 and a vocabulary size of 100,864.

| Hyperparameter | 2.3B Model | 986M Model | 338M Model | 111M Model |
|---|---|---|---|---|
| *Model Architecture* | | | | |
| Total Parameters | 2,321.38M | 985.89M | 338.44M | 110.55M |
| Model Dimension | 2048 | 1536 | 1024 | 512 |
| Number of Layers | 18 | 12 | 8 | 8 |
| Number of Heads | 8 | 8 | 8 | 8 |
| Per Head Dimension | 256 | 256 | 128 | 64 |
| Sequence Length | 2048 | 2048 | 2048 | 2048 |
| *Validation Setup* | | | | |
| Evaluation Batch Size | 1024 | 512 | 128 | 256 |
| Number of Eval Steps | 2 | 4 | 4 | 8 |
| Evaluation Interval | 1000 steps | 1000 steps | 500 steps | 500 steps |

## F    MODEL & EXPERIMENT CONFIGURATIONS

We evaluate cautious weight decay (CWD) across two experimental setups: (1) transformer models ranging from 111M to 2.3B parameters, and (2) the OLMo-1B architecture. All models employ SwiGLU activations and rotary position embeddings (RoPE). To ensure fair comparison, we conduct extensive grid searches to optimize hyperparameters for each baseline optimizer (ADAMW, LION, and MUON) before applying CWD with identical settings. Table 8 details the scaled model configurations, Table 9 presents the OLMo-1B architecture, and the following subsection describes our hyperparameter search methodology.

We conducted an extensive grid search to determine optimal hyperparameters for ADAMW, LION, and MUON optimizers. Our learning rate search employed a quasi-logarithmic grid spanning four orders of magnitude from $1 \times 10^{-5}$ to $1 \times 10^{-1}$, with denser sampling in the critical $10^{-4}$ to $10^{-2}$ range where transformer models typically achieve optimal performance. The grid included standard decade values (e.g., 0.001, 0.01) as well as intermediate points within each logarithmic interval (e.g., 0.2, 0.3, 0.5, 0.8 scaled to each decade) to capture potential performance peaks between order-of-magnitude boundaries, totaling 24 distinct learning rate values. For the learning rate schedule, we systematically evaluated warmup ratios of $\{0, 0.05, 0.1, 0.2, 0.3, 0.4, 0.5\}$, corresponding to 0% to 50% of total training steps dedicated to linear warmup, followed by cosine annealing decay. For ADAMW, we additionally performed a grid search over the momentum parameters $\beta_1$ and $\beta_2$, evaluating combinations of $\beta_1 \in \{0.85, 0.9, 0.95\}$ and $\beta_2 \in \{0.95, 0.98, 0.99, 0.995, 0.999\}$. Our experiments identified $\beta_1 = 0.9$ and $\beta_2 = 0.95$ as the optimal configuration. For LION, we swept $\beta_1 \in \{0.85, 0.9, 0.95\}$ and $\beta_2 \in \{0.95, 0.98, 0.99\}$, finding $\beta_1 = 0.9$ and $\beta_2 = 0.95$ to be optimal. For MUON, we similarly swept momentum coefficients and confirmed 0.95 as optimal.

## G    ADDITIONAL EXPERIMENT RESULTS

This section provides supplementary experimental analyses that further characterize the behavior of cautious weight decay (CWD) across different optimizers and training dynamics. We present detailed visualizations of the mask activation patterns (Figure 7), showing how the fraction of parameters receiving weight decay evolves during training for both ADAMW and MUON optimizers. Additionally, we include comprehensive loss and accuracy curves for all three optimizers (ADAMW, LION, and MUON) across model scales from 111M to 2.3B parameters (Figures 8–10), demonstrating consistent improvements with CWD. Finally, Figure 13 tracks the evolution of parameter norms throughout training, revealing that CWD maintains stable regularization comparable to standard weight decay while achieving superior performance.

Table 9: Model Architecture Configuration for OLMo-1B

| Hyperparameter | Value |
|---|---|
| *Architecture* | |
| Hidden dimension ($d_{\text{model}}$) | 2048 |
| Number of attention heads | 16 |
| Number of layers | 16 |
| MLP ratio | 8 |
| Vocabulary size | 50,280 |
| Embedding size | 50,304 |
| Max sequence length | 2048 |
| *Attention Mechanism* | |
| Positional encoding | RoPE |
| Flash attention | ✓ |
| Multi-query attention | ✗ |
| ALiBi | ✗ |
| Attention dropout | 0.0 |
| Attention layer norm | ✗ |
| *Model Components* | |
| Activation function | SwiGLU |
| Block type | Sequential |
| Weight tying | ✓ |
| Include bias | ✗ |
| Layer norm type | Default |
| Layer norm with affine | ✗ |
| Residual dropout | 0.0 |
| Embedding dropout | 0.0 |
| *Initialization* | |
| Initialization method | Mitchell |
| Initialization device | CUDA |

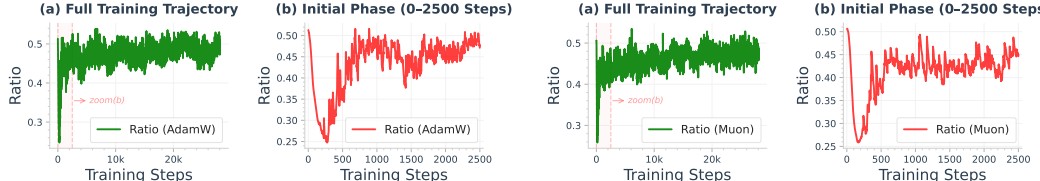

Figure 7: Masked weight-decay activation ratio $r_t := \frac{\|\mathbb{I}(\mathbf{u}_t \mathbf{x}_t > \mathbf{0})\|_1}{d}$, i.e., the fraction of parameters for which the sign-selective mask is active at step $t$ ($d$ = number of parameters). Left: ADAMW; right: MUON. Insets zoom into the first 2.5k steps to highlight early-training behavior. Model: Qwen-0.6B (Yang et al., 2025) trained on The Pile (Gao et al., 2021).

| Accuracy (higher is better) | | | | | | |
|---|---|---|---|---|---|---|
| GPT | ADAMW | | LION | | MUON | |
| Model Size | **Ours** | Base | **Ours** | Base | **Ours** | Base |
| 338M | **0.4232** | 0.4221 | **0.4230** | 0.4211 | **0.4256** | 0.4252 |
| 986M | **0.4566** | 0.4556 | **0.4552** | 0.4545 | **0.4589** | 0.4575 |
| 2B | **0.4847** | 0.4831 | **0.4839** | 0.4830 | **0.4873** | 0.4858 |
| Loss (lower is better) | | | | | | |
| GPT | ADAMW | | LION | | MUON | |
| Model Size | **Ours** | Base | **Ours** | Base | **Ours** | Base |
| 338M | **3.0059** | 3.0136 | **3.0012** | 3.0121 | **2.9851** | 2.9896 |
| 986M | **2.7053** | 2.7142 | **2.7171** | 2.7231 | **2.6873** | 2.6968 |
| 2B | **2.4881** | 2.4973 | **2.4961** | 2.5012 | **2.4703** | 2.4803 |

Table 10: Final evaluation *accuracy* (higher is better) and *loss* (lower is better) comparisons across different model sizes, expanded to the full text width. Our proposed method is benchmarked against three baseline optimizers: ADAMW, LION, and MUON. The best result in each pair is **bolded**.

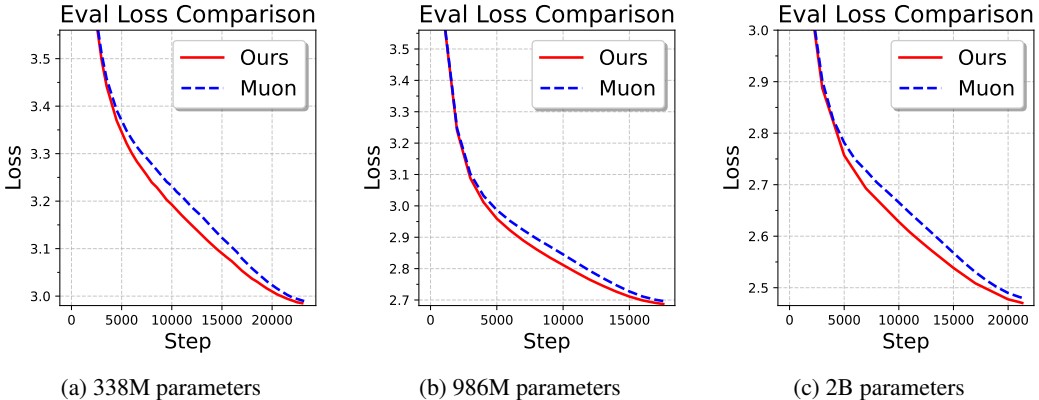

(a) 338M parameters  (b) 986M parameters  (c) 2B parameters

Figure 8: **Training dynamics across model scales with MUON optimizer.** Baseline MUON (dashed) vs. MUON with `CWD` (solid).

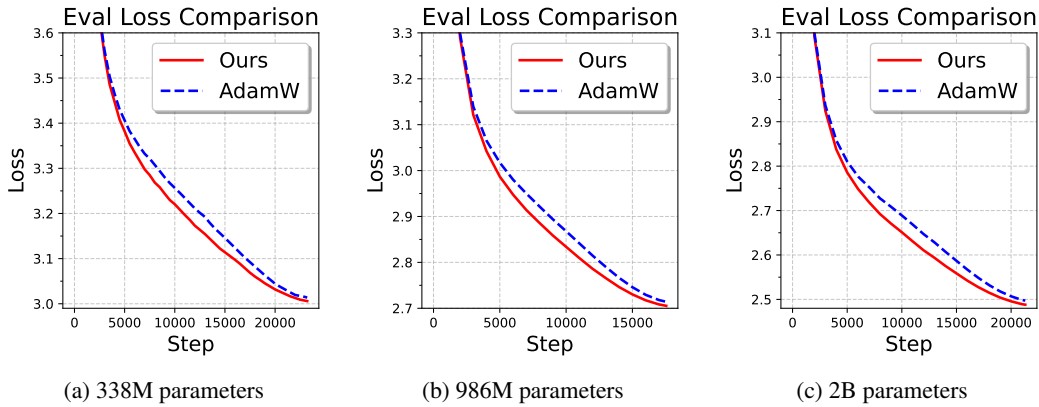

(a) 338M parameters  (b) 986M parameters  (c) 2B parameters

Figure 9: **Training dynamics across model scales with ADAMW optimizer.** We compare baseline ADAMW (dashed) against ADAMW with `CWD` (solid) on models ranging from 338M to 2B parameters.

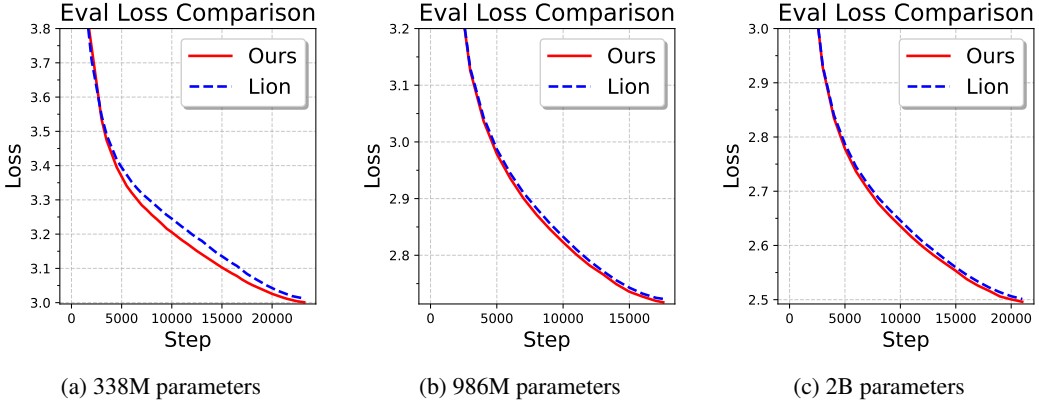

Figure 10: **Training dynamics across model scales with LION optimizer.** Baseline LION (dashed) vs. LION with `CWD` (solid).

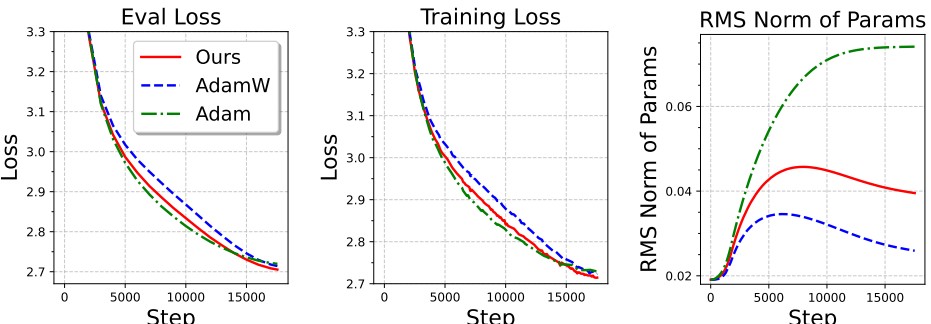

Figure 11: Training dynamics for the 986M-parameter Gemma model.

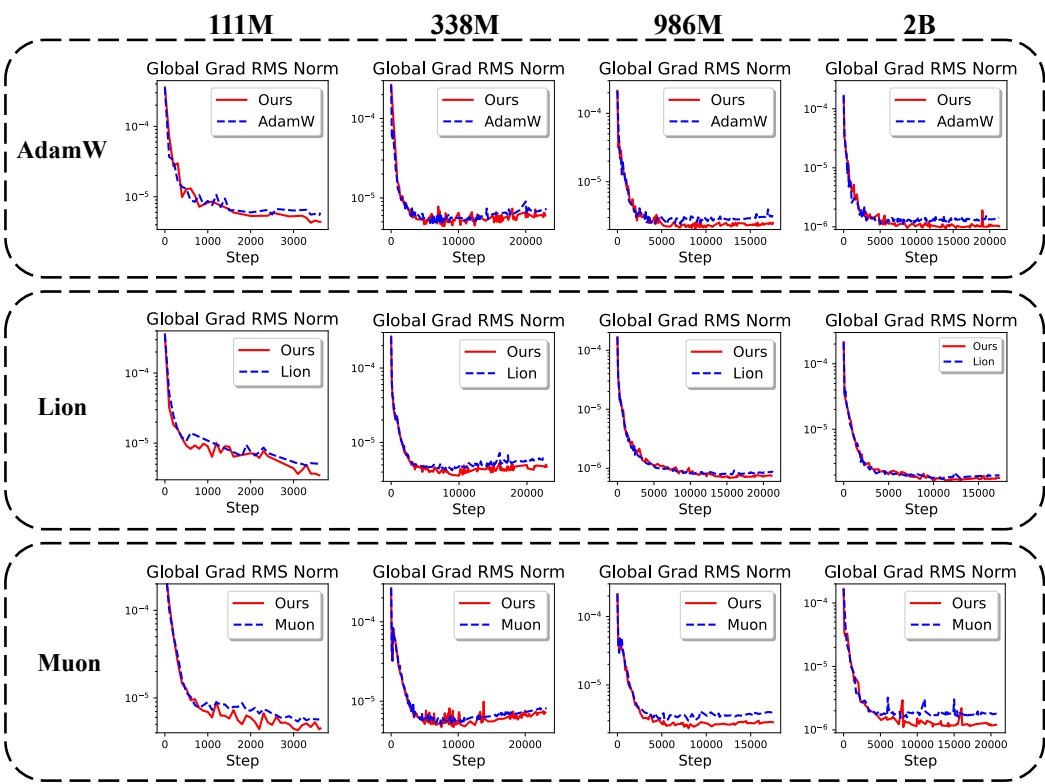

Figure 12: Comparison of gradient norms using RMS normalization across four model sizes: 111M, 338M, 986M, and 2B. All models are trained under Chinchilla settings. `CWD` achieves lower gradient norms across all configurations.

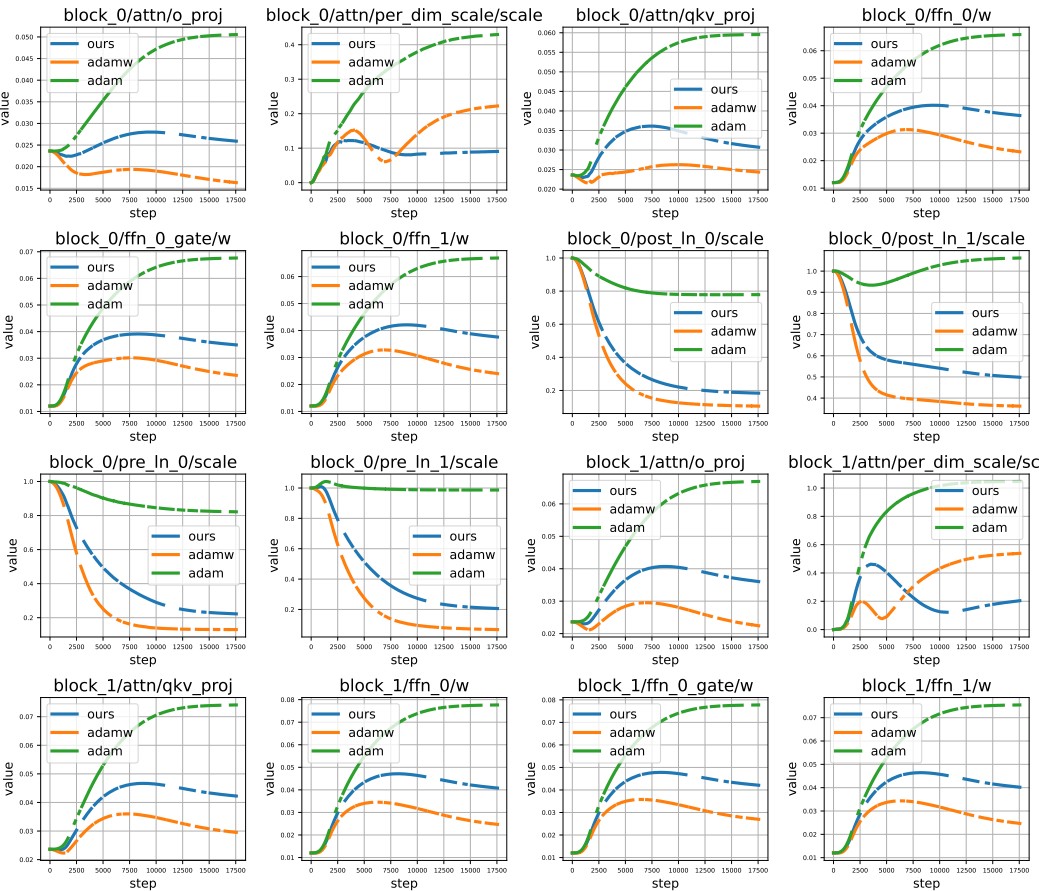

Figure 13: Evolution of parameter norm (RMS) during training for a 986M parameter model. We compare three optimization strategies: ADAMW with weight decay 0.1 (orange), our proposed method (blue), and ADAM without weight decay (green). Our method maintains stable parameter norms comparable to ADAMW while achieving improved performance.

