# OpenReview forum: "Cautious Weight Decay"
_ICLR.cc/2026/Conference — ICLR 2026 Poster_

### Official Review · Reviewer_yQsL · 2025-10-22

**Soundness:** 2
**Presentation:** 3
**Contribution:** 3
**Rating:** 4
**Confidence:** 4

**Summary:**

This paper proposed a novel way of weight decay, which only implements weight decay on the coordinates that the direction aligns with the gradient. The new method is flexible to work with any optimizer and can be implemented easily and efficiently. It can converge to the minimizer manifold, unlike decoupled weight decay will add some additional constraint and can’t reach the minimizer.  The paper theoretically shows the converged point is a Pareto optimal point in the manifold. The experiments demonstrate it empirically works better than decoupled weight decay or no weight decay.

**Strengths:**

1. The theoretical analysis on the continuous part shows the advantage of cautious weight decay, which can be an improvement over decoupled weight decay. The converged point of cautious weight decay has some good implicit bias.
2. The extensive experiment results show the advantage of weight decay is promising.
3. The comparison between different weight decay is very clear with the visualization of trajectories.

**Weaknesses:**

The statement of theorem 1 and the noise assumption are problematic. The four constants $K_1, K_2, K_3, K_4$ should depend on $L, \sigma$ and initial loss gap, which needs to be shown explicitly to check whether the convergence rate is valid. When I read the proof, more problems arise.
1. The noise assumption depends on the batch size, which is unconventional. In other papers that show convergence rate, the variance of the stochastic gradient is fixed $\sigma^2$. Then they can still show $T^{-1/2}$ convergence rate. If this paper’s result is presented with the standard noise assumption, the convergence rate will have a constant term $\sigma$ and therefore is even $\Theta(1)$ for the dependence on $T$.
2. I don’t understand why the iterates are bounded in Lemma 2. Also I don’t know the exact meaning of $G$. If it is a constant independent of $T$ and other problem parameters such as $L$ and $\sigma$, then you are basically cheating with this lemma because you essentially assume the loss function is $G$-Lipschitz rather than $L$-smooth, which is a much stronger assumption. And you also assume the region is bounded, i.e., $||x_t||_\infty \leq R$, which is used in the proof of lemma 6. These should be mentioned explicitly in the main text because they are stronger assumptions than smoothness and noise variance. If $G$ can have some dependence on $T$, then $K_1, K_2, K_3, K_4$ in theorem 1 are not constant and the final convergence rate will have a much worse dependence on $T$.
3. The constants $K_2, K_3, K_4$ are $O(d)$, which means the convergence rate has a very bad dependence on $d$. This is not ideal when $d$ can be much larger than $T$ in reality. Also they are $O(1/\epsilon)$ so the dependence on $\epsilon$ is also not ideal. But this issue is much less serious than the two issues above. It’s good to improve the dependence on $d$ and $\epsilon$ but I can accept the current dependence as long as the proof fixes issue above.

**Questions:**

1. Can you address the issues about convergence rate as I mentioned in weaknesses?
2. I don’t understand the interpretation of the second term in remark 1. Are you suggesting that it will converge to a Pareto front? This claim can only be rigorously shown by a proof similar with Xie and Li 2024.
3. This kind of cautious weight decay makes SGD not rotation-invariant. Have you thought about global cautious weight decay, which implement the weight decay when its inner product with gradient is positive?

There are some possible typos or things that are not clear enough to me:
1. Page 4 line 181: the set of stationary points **is** a union of connected
2. Page 5 line 263: the order comparison between two vectors is a bit ambiguous. I suggest write $y_i$ and $x_i$ explicitly in the scalar order
3. Figure 3: left function should be $f(x,y)=((y-3)^2+(x-3)^2-1)^2$ in the caption. It’s also better to indicate that purpler region is global minimizers.
4. Page 22 line 1153: the last line seems wrong. It seems you are using $||a+b||_2^2 \leq ||a||_2^2 +||b||_2^2$ here. It can be correct if multiplied by 2.

---

> ### Author Response · Authors · 2025-11-18
>
> We thank the reviewer for the time and effort they have invested in meticulously reviewing our work, and we will correct the typos in a revision.
>
> ---
> Before addressing the reviewer’s concerns regarding Theorem 1, we would like to clarify that our main objective is to motivate and introduce CWD, provide a basis for analysis as a starting point to show that it is a theoretically principled technique, and demonstrate its scalability and practical effectiveness on extensive experiments. It is **not** the primary focus of this work to provide an optimal convergence rate for Adam+CWD with the best dependence on $\sigma$, $d$, $\epsilon$, etc. or under the weakest assumptions. We note that many of these questions are not yet well understood even for vanilla Adam or AdamW, and we believe that a full analysis would require substantial work and long term efforts.
>
> For the reviewer’s concerns:
>
> It is important to note that Adam+CWD has a Lyapunov function, which means that the algorithm is essentially well-behaved. The assumptions that we impose here are an artifact of our analysis rather than an inherent limitation of the method.
>
> Weakness 1: Our noise assumption is certainly stronger than the assumption mentioned by the reviewer. However, it is not without precedent in the literature; see e.g. the discussion after Assumption 3 in [BWAA18] and the discussion after Theorem 1 in [ZRS+18], as well as [CLLL24]. Furthermore, there are works that introduce new optimizers without considering the stochastic gradient setting at all in their theoretical results, e.g. [LLH+24, LYL24].
>
> Weakness 2: We first note that $L$-smoothness over a bounded region implies (constant) Lipschitzness over that region. So if we can show that the iterates are bounded by a constant a.s., then we automatically have that the gradients are also bounded by a constant a.s. We will clarify this in a revision.
>
> [BB21] shows that under coercivity (plus standard assumptions), the iterates of Adam are bounded almost surely (Theorem 5.4, with the proof on pages 26–27). Intuitively, this suggests that the iterates of Adam+CWD are also bounded almost surely, since weight decay can only decrease the parameter magnitudes. A formal proof uses a similar approach to [BB21], discretizing our Lyapunov function in Appendix D.4 and using the Robbins–Siegmund theorem to conclude that its value is bounded a.s., which in turn implies the a.s. boundedness of the iterates by the coercivity stated in Assumption 1.
>
> We also remark that many analyses of Adam or Adam-style algorithms assume a priori that the iterates and/or gradients are bounded, e.g. the original Adam paper [KB15] and [RKK18, ZRS+18, CLSH19, CSZL22, DBBU22]. Recent works relaxing these assumptions typically involve long, complicated proofs worthy of a separate paper, e.g. [ZCS+22, LRJ23, HL24].
>
> **We will include this discussion of our assumptions in the main text.**
>
> Weakness 3: See above.
>
> Question 2: This remark is not intended to be a formal proof. It provides intuition for why the second term should show up in the bound at all and how it relates to converging to the Pareto front.
>
> Question 3: Yes! See the discussion with Reviewer XuM6.
>
> # References
>
> [BB21] Anas Barakat, Pascal Bianchi. Convergence and Dynamical Behavior of the ADAM Algorithm for Non-Convex Stochastic Optimization. SIAM Journal on Optimization 31(1):244–274, 2021.
>
> [BWAA18] Jeremy Bernstein et al. signSGD: Compressed Optimisation for Non-Convex Problems. ICML 2018.
>
> [CLLL24] Lizhang Chen et al. Lion Secretly Solves Constrained Optimization: As Lyapunov Predicts. ICLR 2024.
>
> [CLSH19] Xiangyi Chen et al. On the Convergence of A Class of Adam-Type Algorithms for Non-Convex Optimization. ICLR 2019.
>
> [CSZL22] Congliang Chen et al. Towards Practical Adam: Non-Convexity, Convergence Theory, and Mini-Batch Acceleration. Journal of Machine Learning Research 23(229):1–47, 2022.
>
> [DBBU22] Alexandre Défossez et al. A Simple Convergence Proof of Adam and Adagrad. Transactions on Machine Learning Research, 2022.
>
> [HL24] Yusu Hong, Junhong Lin. On Convergence of Adam for Stochastic Optimization under Relaxed Assumptions. NeurIPS 2024.
>
> [KB15] Diederik P. Kingma, Jimmy Ba. Adam: A Method for Stochastic Optimization. ICLR 2015.
>
> [LLH+24] Hong Liu et al. Sophia: A Scalable Stochastic Second-order Optimizer for Language Model Pre-training. ICLR 2024.
>
> [LRJ23] Haochuan Li, Alexander Rakhlin, Ali Jadbabaie. Convergence of Adam Under Relaxed Assumptions. NeurIPS 2023.
>
> [LYL24] Qijun Luo, Hengxu Yu, Xiao Li. BAdam: A Memory Efficient Full Parameter Optimization Method for Large Language Models. NeurIPS 2024.
>
> [RKK18] Sashank J. Reddi, Satyen Kale, Sanjiv Kumar. On the Convergence of Adam and Beyond. ICLR 2018.
>
> [ZCS+22] Yushun Zhang et al. Adam Can Converge Without Any Modification On Update Rules. NeurIPS 2022.
>
> [ZRS+18] Manzil Zaheer et al. Adaptive Methods for Nonconvex Optimization. NeurIPS 2018.

---

> > ### Comment · Reviewer_yQsL · 2025-11-18
> >
> > Thanks for replying to my concerns! To clarify, I really like this paper overall, but I have serious concerns about Section 4. In fact, I might have given a higher score if the original submission had not included the discrete convergence analysis. I understand this is mainly an empirical paper with some theoretical justifications and don’t need an optimal convergence result. I also agree that the current suboptimal result is only an artifact of your analysis and can be fixed using the advanced theoretical tools from related work.
> >
> > However, the way the convergence rate and noise assumptions are presented feels overclaimed and not fully rigorous, which significantly hurt the quality of this paper. I am happy to see that you will discuss about them more clearly and therefore I raised my score. I kindly suggest the authors be precise about the convergence rate and move some of them like the suboptimal noise assumption and stochastic rate to appendix, which is also suggested by reviewer hDqM. There are also some misleading claims in other parts, e.g., line 84 claims standard convergence rate in the smooth nonconvex setting when you actually use coercivity to guarantee gradient boundness and the nonstandard noise assumption.
> >
> > Regarding my question 3, I checked the experiments in the response to Reviewer XuM6. The comparison between SPD and CWD seems promising. However, I believe SPD is different from what I suggest when the inner product mask of SPD $\mathbb{I}(\langle g_t, \theta_{t-1}-\theta_0 \rangle >0)$ and my suggesion $\mathbb{I}(\langle g_t, \theta_{t-1} \rangle >0)$ are different. I’d like to confirm which kind of mask you used in the comparison experiments.

---

> > > ### Author Response · Authors · 2025-11-26
> > >
> > > Regarding question 3, to clarify, we used $\mathbb{I}(\langle\mathbf{g}\_t,\theta_{t-1}-\theta_0\rangle>0)$ as the SPD mask, in accordance with the original formulation of SPD. During the writing of the paper, we also considered the inner product masked mentioned by the reviewer but ultimately chose the elementwise version due to superior empirical performance. For the sake of transparency, we have reproduced some of the results below.
> > >
> > > Results of CWD and SPD on MMLU, AGIEval, and WinoGrande, measured by accuracy.
> > >
> > > | Model       | Method             | MMLU (5-shot)   | AGIEval (3-shot) | WinoGrande (5-shot) |
> > > |------------|--------------------|-----------------|------------------|---------------------|
> > > | TinyLlama  | LoRA               | 25.81 ± 0.07    | 19.82 ± 0.11     | 61.33 ± 0.09        |
> > > | TinyLlama  | SPD                | 26.14 ± 0.08    | **20.21 ± 0.10** | 61.92 ± 0.08        |
> > > | TinyLlama  | Inner product CWD  | 26.02 ± 0.08    | 19.80 ± 0.10     | 61.70 ± 0.09        |
> > > | TinyLlama  | Elementwise CWD    | **26.42 ± 0.09**| 20.12 ± 0.09     | **62.18 ± 0.08**    |
> > > | Mistral-7B | LoRA               | 61.78 ± 0.09    | 27.56 ± 0.07     | 78.85 ± 0.11        |
> > > | Mistral-7B | SPD                | 62.05 ± 0.08    | 27.98 ± 0.06     | 78.81 ± 0.10        |
> > > | Mistral-7B | Inner product CWD  | 61.76 ± 0.09    | 27.90 ± 0.07     | 78.83 ± 0.10        |
> > > | Mistral-7B | Elementwise CWD    | **62.13 ± 0.07**| **28.31 ± 0.06** | **78.92 ± 0.09**    |
> > >
> > > We will make this explicit in the revision and clearly distinguish between the original SPD mask and your proposed alternative when discussing the comparison.
> > >
> > > ---
> > >
> > > Thank you again for the constructive feedback and for raising your score; we are preparing a more detailed reply on the theoretical part to address your concerns, and will revise the paper accordingly to make the analysis more precise and transparent.

---

> > > ### Author Response · Authors · 2025-11-28
> > >
> > > We have taken the reviewer’s suggestions and uploaded a revision. We would also like to make some further clarifications:
> > > - To be very precise, the stochastic gradient assumption that we use, by itself, is **indeed standard**. When $\mathbf{g}_t$ is taken as the mini-batch gradient, its variance is indeed inversely proportional to the batch size. In analyses that do not depend on variable batch sizes, $n\_\text{batch}$ is constant and can therefore be absorbed into $\sigma^2$.
> > > - We acknowledge that requiring $n_\text{batch}=\Theta(T)$, so that the noise term decays at the same rate as the other terms, may be unrealistic in practice. However, as noted in our previous response, this is not without precedent in the literature, and it is conceptually similar to many of the analyses alluded to by the reviewer, where the convergence rate depends on unrealistic values of $\beta_1$, $\eta$, $\lambda$, etc.
> > > - Even with a constant batch size, the convergence rate we present is far from vacuous. For instance, we obtain a $O(T^{-\frac12})$ rate in the deterministic gradient regime. Or if $\frac{\sigma^2}{n_\text{batch}}$ is known to be small, we can still conclude convergence to approximate stationary points.
> > > - If the reviewer is insistent on an $o(1)$ rate with constant batch sizes, we mention the following approach. All of the noise terms, i.e. $\frac{K_4\sigma}{\sqrt{n_\text{batch}}}$, come from (12). If we don’t discard a $\sqrt{1-\beta_1}$ factor, a tighter form of $K_4$ is $\sqrt{\frac{1-\beta_1}{1+\beta_1}}d(G+\epsilon)\left(\frac{G}{\epsilon}+\lambda R\right)$. By setting $\beta_1=\frac{\sqrt{T}-1}{\sqrt{T}+1}$, $K_4$ becomes $O(T^{-\frac14})$. $K_2$ and $K_3$ then become $O(\sqrt{T})$, but this can be balanced by setting $\eta=\Theta(T^{-\frac34})$. The overall convergence rate is $O(T^{-\frac14})$. We would appreciate the reviewer’s thoughts on this.
> > > - We would like to reiterate that analyzing Adam, particularly with weight decay, is challenging. Although significant progress has been made over the past decade, Adam and its variants remain poorly understood from a theoretical standpoint. We consider improving our results in this paper an important direction for future work.

---

### Official Review · Reviewer_hDqM · 2025-10-24

**Soundness:** 4
**Presentation:** 3
**Contribution:** 3
**Rating:** 8
**Confidence:** 4

**Summary:**

The authors propose a simple modification to weight decay, only applying the decay if the optimizer updates and the weights are aligned. The show mathematically that this leads to optimizing the original loss rather than a regularized loss. Through extensive empirical work, the authors demonstrate: 1) Adding the fix to Adam/muon/lion all improves convergence speed for LLM training without retuning hyperparameters. (fig4) 2) The performance is consistently better across levels of weight decay (fig1), 3) ImageNet accuracy is also improved.

**Strengths:**

- The method is very simple
- The empirical results are very convincing, especially since they use multiple codebases
- The paper is well written and easy to follow

**Weaknesses:**

- Personally, I think the theoretical results are distracting. I'd prefer that they be moved to the appendix and that the main paper is focused on the empirical results. That is what most people will care about.
- There is some missing related work (e.g., "Understanding decoupled and early weight decay")

**Questions:**

1. Are you able to provide scaling plots of final-loss-vs-model size for the baseline and your method? It would be interesting to see if the "boost" in performance is retained as the model is scaled.

---

> ### Author Response · Authors · 2025-11-18
>
> We thank the reviewer for their thorough evaluation and constructive feedback.
>
> ---
> Regarding the weaknesses,
> - We thank the reviewer for this suggestion. We will consider balancing the theory and other parts in a revision, further emphasizing the algorithm and the empirical advantages of our method.
> - We thank the reviewer for mentioning [BWG21]. We will cite this work, along with the works mentioned by other reviewers, in a revision.
>
> To answer the review’s question, [this figure](https://anonymous.4open.science/r/cautious_weight_decay_iclr2026-1D75/scaling_loss_vs_model_size%20(1).pdf) shows how the final validation loss scales with model size for two optimizers: a baseline method (AdamW) and our proposed method ("Ours"). We train models with 111M, 338M, 986M, and 2B parameters on the same data and report the final validation loss after convergence. As model size increases, both methods benefit from lower loss, but our method consistently achieves a lower loss than the baseline at every scale, and the gap remains stable or slightly widens as parameters grow, indicating that the proposed optimizer continues to provide benefits in the large-model regime.
>
> # References
> [BWG21] Johan Bjorck, Kilian Weinberger, Carla Gomes. Understanding Decoupled and Early Weight Decay. AAAI 2021.

---

> > ### Author Response · Authors · 2025-11-26
> >
> > > We would like to briefly follow up on our earlier author response to this review. In that response, we addressed the main points raised in the “Weaknesses” and “Questions” sections:
> > >
> > > * **Balance between theory and empirical results.**
> > >   We clarified that, in a revision, we plan to streamline the theoretical exposition (moving some technical material to the appendix) and place stronger emphasis in the main text on the algorithmic idea and empirical advantages of cautious weight decay.
> > >
> > > * **Related work.**
> > >   We acknowledged the omission of *“Understanding Decoupled and Early Weight Decay”* [BWG21] and committed to citing and positioning our contribution relative to this work, as well as other related work highlighted by the reviewers.
> > >
> > > * **Scaling behavior.**
> > >   In response to the question on scaling, we ran additional experiments measuring how the final validation loss scales with model size for AdamW vs our method (“Ours”) at 111M, 338M, 986M, and 2B parameters on the same dataset. Across all these scales, our method achieves consistently lower final validation loss, and the gap remains stable or slightly increases with model size, indicating that the benefits of cautious weight decay persist in the large-model regime.
> > >
> > > We would be grateful if the reviewer could let us know whether these clarifications and additional results address their concerns, especially regarding the scaling question. If there are remaining issues or if further plots/ablations would be helpful (within the constraints of the discussion period), we are very happy to provide additional details or clarification.

---

### Official Review · Reviewer_Jtg5 · 2025-10-30

**Soundness:** 4
**Presentation:** 4
**Contribution:** 4
**Rating:** 10
**Confidence:** 4

**Summary:**

Cautious weight decay describes a one-line modification of weight decay such that it is applied only on parameters that are aligned with the update direction.
The paper shows that this is related to minimizing the unregularized loss, with limiting point such that all weights have minimal magnitude locally.
Large-scale experiments show the effectiveness of the method.

**Overall,** the contributions and evaluations in this paper are of highest quality and the presented technique is likely to be adapted in practice quickly.

**Strengths:**

This paper presents a seemingly simple idea that can be implemented with changing one line of code change. The effects of this modification are extremely well explained, and justified through experiments on industry-scale problems for language model training; the method also performs well on standard classification models for Imagenet. Several ablations are provided that justify the exact construction of cautious weight decay.

Besides the clarity and convincing experiments, the paper also contains convergence analysis of the method, and extensive motivation and background through Lyapunov analysis.

**Weaknesses:**

The paper has no major weaknesses. The only minor point/question is on the motivation of CWD through Lyapunov analysis: the design of CWD is not explicitly reflected in the choice of its Lyapunov function. This leaves open the question whether other modifications of weight decay would allow for the same Lyapunov function, and if these alternatives would be interesting/competitive methods. From the current presentation, I could not find an argument why this modification of weight decay would be the singular one to yield improvements.

**Questions:**

* Did you test the effect of CWD for different learning-rate schedules (e.g. cosine and WSD)? Did you test whether CWD is compatible with AdamC by Defazio, 2025, and if there is a compound improvement of using both?
* Does the floating-point format have an effect on CWD, in the sense that using low-precision formats might introduce a harmful bias in the direction/weights?
* Figure 3 left: why does the CWD trajectory not stop once it reaches the Pareto set?
* Assumption 1 states that the iterates are bounded due to coercivity. Can you provide a short proof for this, and also clarify in which probabilistic sense (as the iterates are random variables)?

Minor:

* line 220: why does $m$ necessarily decay to zero? In the stochastic case, with non-vanishing noise, this is not clear to me.
* The entire paper uses the notation $(\nabla f(x_t) x_t)^+$, but this is missing the transposition operator on the gradient.

---

> ### Author Response · Authors · 2025-11-18
>
> We thank the reviewer for their thorough evaluation and constructive feedback.
>
> ---
> Regarding the motivation of CWD through Lyapunov analysis: This is an interesting question. Remark 3 in Appendix D.4 motivates CWD from a Lyapunov perspective. It is possible that there are alternative modifications that could lead to other methods, but we leave this to future work.
>
> For the questions,
> - We observe improved performance for both cosine scheduling and the Warmup–Stable–Decay (WSD) schedule, which uses a 10% warmup, a long stable phase, and a 20% final decay.
>
> | Model Size | Optimizer        | Scheduler | Final Validation Loss |
> |------------|------------------|-----------|-----------------------|
> | 338M       | AdamW (baseline) | Cosine    | 3.0136                |
> | 338M       | AdamW (baseline) | WSD       | 3.0101                |
> | 338M       | Cautious AdamW   | Cosine    | **3.0059**                |
> | 338M       | Cautious AdamW   | WSD       | **3.0014**                |
>
> We thank the reviewer for mentioning [Def25]. We love this work a lot, and we have observed a compound improvement on pretraining using Gemma-based model architectures.
> | Model Size | Optimizer                      | Final Validation Loss |
> |------------|--------------------------------|-----------------------|
> | 338M       | AdamW (baseline)              | 3.0136                |
> | 338M       | AdamW + Cautious Weight Decay | 3.0059                |
> | 338M       | AdamC                         | 3.0087                |
> | 338M       | AdamC + Cautious Weight Decay | **2.9915**                |
> - Our experiments use fp32 for optimizer states and bf16 for model parameters. We ask the reviewer to clarify what they mean by low-precision formats.
> - The exact point that an optimizer with CWD converges to is dependent on, among other things, the base optimizer. Here, Adam+CWD actually converges to the minimum $\ell_\infty$ norm minimizer, which is in general a Pareto optimal point. Hence, the trajectory might still move within the Pareto set, but it will enter the set in general and remain there.
> - Thanks for pointing this out. We mean that the iterates are bounded almost surely. See the discussion with Reviewer yQsL for details.
> - Here, we work in the continuous-time ODE limit, in which there is no gradient noise. This simplified setting makes it cleaner to study the underlying behavior of optimizers with CWD.
> - As mentioned in the footnote on page 3, we use $\nabla f(\mathbf{x}_t)\mathbf{x}_t$ as a shorthand for the elementwise product of $\nabla f(\mathbf{x}_t)$ and $\mathbf{x}_t$.
>
> # References
>
> [Def25] Aaron Defazio. Why Gradients Rapidly Increase Near the End of Training. CoRR, abs/2506.02285, 2025.

---

> > ### Comment · Reviewer_Jtg5 · 2025-11-26
> > **Acknowledgement**
> >
> > Dear authors,
> >
> > thank you for your response. Regarding the theoretical results, I support the view of Reviewer yQsL that the presentation needs to be clarified and mathematically rigorous. However, the main contribution of this paper seems to be the empirical results, thus I keep my positive evaluation.

---

### Official Review · Reviewer_XuM6 · 2025-11-01

**Soundness:** 3
**Presentation:** 4
**Contribution:** 2
**Rating:** 6
**Confidence:** 4

**Summary:**

This paper proposes Cautious Weight Decay (CWD) — a simple, optimizer-agnostic modification that applies weight decay only when the update direction matches the parameter sign. Unlike standard weight decay, CWD preserves the original objective and admits a bilevel/sliding-mode interpretation. It requires no extra hyperparameters and works as a drop-in replacement for AdamW, Lion, and Muon. Experiments on large-scale language model pretraining and ImageNet classification show consistent improvements in loss and accuracy.

**Strengths:**

1. Well written, easy to follow, and clearly presented.
2. Theoretical foundations are solid and supported by extensive experiments across both vision and language tasks.

**Weaknesses:**

While the proposed method is elegant and broadly applicable, its novelty may be somewhat limited. Conceptually, it appears closely related to Selective Projection Decay (SPD) [1], although applied in a different setting. SPD selectively applies regularization to layers whose updates are inconsistent, determined by the sign of the inner product between the negative gradient direction $(-g_t)$ and the accumulated parameter change $(\theta_{t-1} - \theta_0)$. A positive inner product indicates progress toward lower loss, whereas a negative value implies potential movement toward higher loss, prompting SPD to impose stronger penalties on such layers. The main difference is that SPD is designed for robust fine-tuning and therefore regularizes the $L_2$ distance between pretrained and finetuned weights $(\theta_{t-1} - \theta_0)$, while CWD regularizes the $L_2$ norm of the weights themselves $(\theta_{t-1})$. Under this substitution, the core mechanism becomes mathematically similar, making the originality of CWD somewhat questionable, despite the fact that this paper provides stronger theoretical justification and demonstrates wider applicability across optimizers and domains.

[1] Tian, Junjiao, Chengyue Huang, and Zsolt Kira. "Rethinking weight decay for robust fine-tuning of foundation models." Advances in Neural Information Processing Systems 37 (2024): 22418-22440.

**Questions:**

Please refer to the weaknesses.

---

> ### Author Response · Authors · 2025-11-18
>
> We thank the reviewer for their reviewing efforts and for pointing us to the relevant work [THK24]. We really like this work [THK24]. We will certainly cite this work and include a detailed comparison with our contributions in a revision.
>
> ---
> We argue that the technique proposed by [THK24], SPD, appears similar to CWD but differs from it in several important ways, including the intended setting, empirical performance, and theoretical behavior.
>
> 1. As mentioned by the reviewer, the two methods are applied in different settings. SPD is intended for fine-tuning, while CWD is a general optimizer technique.
> 2. The inner product mask $\mathbb{I}(\langle\mathbf{g}\_t,\theta_{t-1}-\theta_0\rangle>0)$ used by SPD is different from and performs worse than the elementwise mask $\mathbb{I}(\mathbf{u}_t\odot\mathbf{x}_t\geq\mathbf{0})$ used by CWD. The former is a scalar that controls the application of weight decay, while the latter is a vector that is multiplied entrywise with the weight decay term.
>
> Furthermore, one can intuitively expect that a global inner product mask would result in slower convergence than an elementwise mask, since weight decay is activated in an all-or-nothing fashion in the former.
>
> Indeed, we verified this with several experiments comparing CWD and SPD. We evaluate our methods on an instruction-following fine-tuning task using the Alpaca GPT-4 dataset, which contains 52k instruction–response conversation pairs generated by GPT-4. The effectiveness of fine-tuning is then assessed on three benchmarks: MMLU, comprising 57 tasks and 14,079 questions covering a broad range of world knowledge; AGIEval, a human-centric benchmark with 9,316 instances targeting general reasoning and problem-solving skills; and WinoGrande, a large-scale commonsense reasoning dataset with 44,000 instances designed to challenge models’ understanding of context and everyday knowledge. We assessed two baseline models, TinyLlama [ZZWL24] and Mistral-7B [JSM+23].
>
> Results of CWD and SPD on MMLU, AGIEval, and WinoGrande, measured by accuracy.
> | Model       | Method | MMLU (5-shot)   | AGIEval (3-shot) | WinoGrande (5-shot) |
> |------------|--------|-----------------|------------------|---------------------|
> | TinyLlama  | LoRA   | 25.81 ± 0.07    | 19.82 ± 0.11     | 61.33 ± 0.09        |
> | TinyLlama  | SPD    | 26.14 ± 0.08    | **20.21 ± 0.10**     | 61.92 ± 0.08        |
> | TinyLlama  | CWD    | **26.42 ± 0.09**    | 20.12 ± 0.09     | **62.18 ± 0.08**        |
> | Mistral-7B | LoRA   | 61.78 ± 0.09    | 27.56 ± 0.07     | 78.85 ± 0.11        |
> | Mistral-7B | SPD    | 62.05 ± 0.08    | 27.98 ± 0.06     | 78.81 ± 0.10        |
> | Mistral-7B | CWD    | **62.13 ± 0.07**    | **28.31 ± 0.06**     | **78.92 ± 0.09**        |
>
> We can see that there is a clear benefit by using SPD compared to the baseline LoRA, CWD also performs well.
>
> 3. It is not obvious if SPD exhibits the theoretical properties of CWD. Our theoretical insights and results for CWD comprise a significant theoretical contribution in their own right.
>
> # References
>
> [THK24] Junjiao Tian, Chengyue Huang, Zsolt Kira. Rethinking Weight Decay for Robust Fine-Tuning of Foundation Models. NeurIPS 2024.
>
> [ZZWL24] Peiyuan Zhang, Guangtao Zeng, Tianduo Wang, Wei Lu. TinyLlama: An Open-Source Small Language Model. CoRR, abs/2401.02385, 2024.
>
> [JSM+23] Albert Q. Jiang et al. Mistral 7B. CoRR, abs/2310.06825, 2023.

---

> ### Author Response · Authors · 2025-11-26
>
> We thank the reviewer for pointing out the connection to SPD [THK24] and for their constructive feedback. We will cite [THK24] and add a dedicated comparison in the revised version.
>
> ---
>
> At a high level, SPD and Cautious Weight Decay (CWD) share the intuition that regularization should depend on the alignment between an update direction and a reference direction. However, their goals, mechanisms, and theoretical properties differ in important ways.
>
> ---
>
>
> ### Intended setting.
>
> SPD is explicitly designed for **robust fine-tuning** of pretrained models, where the regularizer penalizes the distance between finetuned and pretrained weights. In contrast, **CWD is a generic modification of weight decay that regularizes the weight norm itself** and is designed to be a **drop-in, optimizer-agnostic technique** that applies to
>
> - large-scale pretraining,
> - standard supervised training (e.g., ImageNet), and
> - fine-tuning.
>
> This difference is crucial for our empirical scope: **our main results are in pretraining and classification regimes that are outside of SPD’s stated focus**.
>
> ---
>
> ### Mechanism.
>
>
> Unlike SPD, which gates regularization using a **layerwise scalar** derived from the inner product between the **gradient and the parameters** (or their displacement from the pretrained weights), **CWD relies on an elementwise interaction between the optimizer update and the parameter**, yielding a **coordinate-wise mask** that is applied entrywise to the weight decay term. This is more than an implementation detail:
>
> - It allows CWD to react at the level of individual parameters rather than entire layers, which we observe leads to more aggressive yet stable progress.
> - It avoids the “all-or-nothing” behavior of a global scalar gate, which can slow convergence because aligned and misaligned coordinates are treated identically whenever the scalar is active.
>
> ---
>
> ### Empirical comparison.
>
>
> We verified this difference on an instruction-following fine-tuning setup (Alpaca GPT-4) and three benchmarks (MMLU, AGIEval, WinoGrande) using TinyLlama [ZZWL24] and Mistral-7B [JSM+23] as shown in the table above. SPD consistently improves over a LoRA baseline, confirming the benefit of sign-aware regularization, while **CWD is at least as good and often better**:
>
> - For TinyLlama, CWD improves on SPD on MMLU and WinoGrande.
> - For Mistral-7B, CWD matches or exceeds SPD on all three benchmarks.
>
> These results complement our main experiments and show that **the CWD mechanism is competitive even in SPD’s target fine-tuning setting, while also being effective in pretraining and vision.**
>
> ---
>
> ### Theoretical properties.
>
>
> Our main theoretical contributions are specific to CWD as a form of cautious weight decay:
>
> - **CWD preserves the original objective** while still inducing an effective regularization behavior.
> - **It admits a bilevel / sliding-mode interpretation** that characterizes the limiting dynamics of the parameters.
> - **It applies to a broad class of optimizers (AdamW, Lion, Muon) without additional hyperparameters.**
>
> To the best of our knowledge, SPD has not been analyzed from this perspective, and it is not clear whether the same properties hold for its inner-product-based, layerwise mechanism. We will clarify this point in the revision by explicitly stating which results are unique to CWD.
>
> ---
>
>
> ### Summary.
> In summary, while SPD and CWD share a common high-level intuition, CWD contributes
>
> - **a different and more fine-grained mechanism** (elementwise sign-based mask vs layerwise scalar mask),
> - **a broader empirical scope** (pretraining, supervised vision, and fine-tuning, across multiple optimizers), and
> - **new theoretical guarantees** that, to our knowledge, have not been established for SPD.
>
> We will revise the paper to clearly articulate these distinctions and to present **CWD as a unifying, optimizer-agnostic formulation of sign-aware weight decay rather than a restatement of SPD**.

---

> > ### Author Response · Authors · 2025-11-27
> >
> > We thank the reviewer again for their thoughtful feedback. In the revision, we have added a dedicated comparison to SPD, clarified the distinctions in setting, mechanism, and theory, and expanded the empirical discussion accordingly.
> >
> > We hope these updates satisfactorily address your concerns. If so, we would kindly appreciate your consideration in updating the evaluation. We are of course happy to clarify anything further if needed.

---

### Author Response · Authors · 2025-11-28
**General Response**

We sincerely thank all of the reviewers for their helpful feedback and their commitment to improving the quality of our paper. We have uploaded a revision that clarifies the theoretical section and substantially expands the experimental section (Section 4) to better support our claims and address the reviewers’ concerns. The changes are highlighted in the PDF and summarized below.

**Scaling with model size.** We added experiments measuring how final validation loss scales with model size for AdamW vs our method (“Ours”) at 111M, 338M, 986M, and 2B parameters on the same dataset. Across all scales, our method achieves consistently lower final validation loss, and the gap remains stable or slightly increases with model size, indicating that the benefits of cautious weight decay persist in the large-model regime.

**Compound improvement with other techniques.** On Gemma-based architectures at 338M parameters, we compared AdamW, AdamC, and their variants with Cautious Weight Decay. CWD improves both AdamW and AdamC, and the combination AdamC + CWD yields the lowest final validation loss, showing that our method provides compounding gains when paired with stronger optimizers.

**Robustness to learning-rate schedules.** We also evaluated cosine and Warmup–Stable–Decay (WSD) schedules. Adam + CWD outperforms standard AdamW under both schedules, with the best performance obtained by Adam + CWD with WSD. This demonstrates that the benefits of CWD are not tied to a specific scheduler.

**Additional ablations.** We ran instruction-following fine-tuning experiments on Alpaca GPT-4 with TinyLlama and Mistral-7B, evaluating MMLU, AGIEval, and WinoGrande. We compared LoRA, SPD, a layerwise “inner-product CWD” variant, and our proposed “elementwise CWD”. Both SPD and CWD improve over LoRA, while elementwise CWD matches or outperforms SPD and the inner-product variant on most benchmarks. This directly addresses the connection to SPD and isolates the benefit of our elementwise design.

**Discrete-time analysis.** At the suggestion of Reviewers hDqM and yQsL, we moved the discrete-time analysis to Appendix E. For the sake of clarity, we removed the assumption of coerciveness and added explicit bounded iterate and bounded gradient assumptions along with a justification. **The statement and proof of Theorem 1 are fully mathematically rigorous.** We also do not make misleading claims about this result anywhere in the paper.

**Additional discussion of related work.** We have cited additional related work, including the works mentioned by the reviewers [THK24, BWG21, Def25], and expanded the related work section (Section 5).

We also corrected minor typos.

---

### Meta-Review · Area_Chair_Q5P7 · 2026-01-07

**Summary:**

This paper proposes a simple modification of weight decay, named Cautious Weight Decay (CWD), that applies weight decay only to parameter coordinates whose signs align with the optimizer update direction. The authors argue that, unlike standard decoupled weight decay, which implicitly optimizes a constrained objective, CWD preserves the original unregularized objective and can be interpreted via bilevel dynamics upon reaching the stationary manifold.

**Reviewer Concerns:**

This paper has mixed initial opinions, with two strong positive opinions and two borderline opinions. The main strengths are (1) the method is simple and practical, and (2) the comprehensive experiments.

The key concerns raised by the reviewers can be summarized by (1) lack of novelty compared to previous works (e.g., SPD) (XuM6) and (2) somewhat insufficient support for the discrete-time convergence analysis (Theorem 1) (yQsL, hDqM, Jtg5).

The first concern was addressed by the additional comparison experiment with SPD, an inner-product variant, and elementwise CWD. I think that the rebuttal comment clarifies the difference between CWD and SPD well.

The second concern was partially addressed in the revised paper. The revised paper moved the discrete-time analysis from the main manuscript to the appendix and clarified the assumptions. Although the revised paper reduces the risk of misunderstanding and misinterpretation, the core concern about strong assumptions and how readers interpret the discrete-time guarantees is only partially resolved. There were also additional concerns, such as scaling, generalizability to other optimizers, and robustness to the choice of the lr scheduler. I think that they are well addressed by the rebuttal comment.

Discrete-time analysis. At the suggestion of Reviewers hDqM and yQsL, we moved the discrete-time analysis to Appendix E. For the sake of clarity, we removed the assumption of coerciveness and added explicit bounded iterate and bounded gradient assumptions along with a justification. The statement and proof of Theorem 1 are fully mathematically rigorous. We also do not make misleading claims about this result anywhere in the paper.

Overall, I think that the advantages of this paper (simplicity and strong empirical results) outweigh its disadvantages (e.g., not fully resolved concerns on the theoretical results). Hence, I recommend acceptance of this paper.

**Reviewer Scores:**

I don't think the reviewers who have strong initial opinions bump down their opinions after discussion. The reviewers who have borderline initial opinions might be able to change their opinions, but in my opinion, the concerns raised by yQsL might not be fully resolved by the rebuttal comment. However, I think that yQsL might change their opinion slightly above the borderline if they have sufficient discussions between the authors.

---

### Decision · Program_Chairs · 2026-01-26

Accept (Poster)